# TREM2 expression level is critical for microglial state, metabolic capacity and efficacy of TREM2 agonism

Astrid F. Feiten [1,2,18], Kilian Dahm [3,4,5,18], Kai Schlepckow [2,18], Bettina van Lengerich[6], Jung H. Suh [6], Anika Reifschneider[1], Benedikt Wefers [2], Laura M. Bartos[7], Karin Wind-Mark [7], Lis de Weerd [2], Thomas Ulas[3,5,8], Elena De-Domenico [3,5,8], Pia Grundschöttel[3,5,8], Stefan Paulusch[3,5,8], Benjamin Tast[9], Tamisa Honda [9], Stephan A. Müller [2,10], Matthias Becker [3,11], Igor Khalin [12,13], Alessio Ricci [12], Arthur Liesz [12,14], Bettina Brunner[2], Claudia Krenner[1], Katrin Buschmann[1], Brigitte Nuscher[1], Lena Spieth[2,15], Niklas Junker[2,15], Stefan A. Berghoff[2,15], Sonnet S. Davis[6], Jonas J. Neher [1,2,14], Wolfgang Wurst [2,16,17], Nikolaus Plesnila [12,14], Joseph W. Lewcock [6], Mikael Simons [2,12,14,15], Stefan F. Lichtenthaler [2,10,14], Gilbert Di Paolo [6], Matthias Brendel [2,7,14], Anja Capell [1,2], Kathryn M. Monroe [6,19] ✉, Joachim L. Schultze [3,5,8,19] ✉ & Christian Haass [1,2,14,19] ✉

Triggering receptor expressed on myeloid cells 2 (TREM2) is a central regulator of microglial activity and loss-of-function coding variants are major risk factors for late onset Alzheimer's disease (LOAD). To better understand the molecular and functional changes associated with TREM2 signalling in microglia, we generated a TREM2 reporter mouse. In *APP* transgenic animals, bulk RNA-sequencing of isolated microglia sorted based on reporter expression highlighted TREM2 level-related changes in major immunometabolic pathways, and enrichment of genes in oxidative phosphorylation and cholesterol metabolism in microglia with increased TREM2 expression. Metabolic and lipidomic profiling of sorted microglia showed that, independent of Aβ pathology, TREM2 expression correlated with signatures consistent with increased cellular redox, energetics, and cholesterol homoeostasis. In accordance, metabolic activity correlated with phagocytic capacity. Finally, we performed chronic treatment with a TREM2 agonist antibody and identified a window of TREM2 expression where microglia are most responsive, thereby informing clinical applications of TREM2 agonists.

Although the original descriptions of Alzheimer's disease (AD) pathology already suggested glial pathology[1], its contribution to disease progression remained largely controversial until genome wide association studies (GWAS) identified risk loci for late onset AD (LOAD) expressed predominantly in microglia[2]. Now microglia are known to form a barrier around amyloid plaques[3–6]. These plaque-associated microglia are phagocytic and mediate removal of aggregated amyloid β-peptide (Aβ)[7]. Furthermore, formation of this barrier restricts the

amount of toxic Aβ oligomers that can escape from the plaque and thereby may limit neuritic dystrophy and neurotoxicity[4,5]. In order to perform protective functions, microglia must change their activation state[8]. A major regulator of the transition from homoeostatic microglia to disease-associated microglia (DAM) is the triggering receptor expressed on myeloid cells 2 (TREM2)[8–10]. Heterozygous coding variants of this transmembrane receptor have been associated with LOAD[11,12], while homozygous loss-of-function mutations in TREM2 or its signalling adaptor DAP12 (encoded by the gene *TYROBP*) are causative for Nasu-Hakola disease (NHD), an early onset dementia with bone cysts[13]. In addition, mono allelic deletions of *TYROBP* were recently identified as a risk factor for late onset AD, further supporting dysfunction of TREM2 signalling as a trigger for LOAD[14]. Within the brain, TREM2 is predominantly expressed in microglia[15], where it forms a complex with DAP12 or DAP10 (encoded by the gene *HCST*) to initiate downstream signalling mediated by spleen tyrosine kinase (SYK)-dependent or independent pathways[16–19]Activation of TREM2 has been associated with functions such as phagocytosis[9,15,20], synaptic pruning[21], and controlling the inflammatory state through cytokine and chemokine release[15,22]. It has also been suggested that TREM2 can regulate tau kinases[23], glucose uptake[24–26], and microglial metabolism[24,26–28], linking it to core AD pathologies. Furthermore, AD patients with underlying *TREM2* loss-of-function variants and *Trem2* knockout (KO) mice show a reduction of microglial clustering around amyloid plaques[6,29] and a downregulation of mammalian target of rapamycin (mTOR), which plays an important role in cellular metabolism[27]. Microglial TREM2 has been shown to be vital in clearing myelin debris as the induction of a DAM state seems to be required to efficiently clear cholesteryl esters derived from myelin cholesterol[30,31]. In the periphery, TREM2 is a master regulator of lipid-associated macrophages (LAM)[32]. In adipose tissue, loss of TREM2 leads to systemic hypercholesterolaemia, inflammation and glucose intolerance, suggesting a critical role for TREM2 in energy metabolism and lipid catabolism[32].

Recently, agonistic antibodies targeting TREM2 have been explored as treatment options for LOAD[33,34]. Agonistic antibodies stimulate microglial proliferation, mitochondrial metabolism and glucose uptake[26,34,35] and some of them reduce amyloid plaque load[36]. The very first phase II clinical trial (INVOKE-2) by Alector failed since none of the primary and secondary endpoints were reached and amyloid-related imaging abnormalities (ARIA) precluded continuation[37]. Additional clinical trials with TREM2 agonists are still in early phases[34]. The detrimental outcome of INVOKE-2 together with numerous failed clinical trials using anti-amyloid immunotherapy should be taken as a serious warning to test newly developed medications in patients too early without having in depth knowledge of the targeted pathway, including the best timing for treatment. For TREM2 agonistic therapeutic interventions, clinicians need to know the state of microglia activation, which allows for optimal stimulation of TREM2-dependent protective functions.

To address these pivotal questions, we generated a TREM2 reporter mouse model which allows to decipher how TREM2 expression levels in amyloid-burdened and healthy mice affect metabolic functions of microglia and govern responses to a TREM2 agonist antibody. We isolated microglia and performed cell sorting based on the expression of the fluorescent TREM2 reporter, which allowed for analyses of subpopulations of microglia expressing low, mid, and high TREM2 from the same mouse. Notably, we found that the amount of TREM2 directly relates to glucose uptake levels and is an important regulator of microglia metabolism, independent of disease context. The metabolic and lipidomic profile of high TREM2 expressing microglia was shifted towards increased phagocytosis, increased energetic and metabolic capacity, and decreased cholesterol, suggesting a protective microglial change induced by robust TREM2 activation. Moreover, treatment with an agonistic TREM2 antibody

revealed mid TREM2 expressing cells to be most responsive. These findings confirm TREM2 as a crucial and dose-dependent regulator of microglial metabolism which has important implications for the design of clinical trials.

## Results

### Generation of a TREM2 reporter mouse reveals gradual upregulation of TREM2 in microglia

We generated a TREM2 reporter mouse line to investigate how differences in TREM2 expression and its functional consequences correlate with amyloid plaque pathology. We used the fluorophore mKate2[38] as a reporter, separated from *Trem2* by the P2A sequence to express both proteins individually under the control of the same promotor[39]. To ensure that the mKate2 reporter protein is retained within the cell, we attached the endoplasmic reticulum (ER)-retention sequence 'KDEL'[40] to its C-terminus (Fig. 1A). We first confirmed the expression of both proteins TREM2 and mKate2 from the polycistronic cDNA construct in HeLa cells (Fig. 1B). mKate2 was expressed evenly throughout the endoplasmic reticulum and clearly separated from cell surface TREM2 staining (Fig. 1B). Immunoblot analyses of cell lysates upon transient transfection of the *TREM2-P2A-mKate2-KDEL* cDNA in HeLa cells revealed that both TREM2 and mKate2 were degraded in a time-dependent manner (Fig. 1C). Quantification indicates that mKate2 is degraded more slowly compared to TREM2 (Fig. 1D), which is consistent with rapid proteolytic shedding of TREM2 on the cell surface[41]. Thus, mKate2 reflects TREM2 expression, even in cells that are no longer TREM2 signalling competent due to continuous shedding of the cell surface receptor. Using CRISPR/Cas9, we next introduced the P2A-mKate2-KDEL cassette at the 3' end of exon 5 into the mouse *Trem2* locus (Fig. 1E) and further analysed homozygous (*Trem2-mKate2*[KI/KI], hereafter *Trem2*[KI/KI]), heterozygous (*Trem2-mKate2*[KI/wt], hereafter *Trem2*[KI/wt]), and wild type (*Trem2-mKate2*[wt/wt], hereafter *Trem2*[wt/wt]) mice. Co-staining of mKate2 and the microglial marker Iba1 showed that mKate2 expression localised to microglia (Fig. 1F). Insertion of the mKate2 cassette did not significantly alter total amounts of *Trem2* mRNA (Supplementary Fig. 1A). As expected, a gene dose dependent increase of *mKate2* mRNA was observed (Supplementary Fig. 1B), with a corresponding decrease of the unmodified (wt) *Trem2* mRNA, i.e., mRNA lacking the silent mutations introduced during generation of the reporter line (Supplementary Fig. 1C). Next, we confirmed microglial activation as a proxy for TREM2 receptor function in the reporter mice using a well-established model for controlled cortical impact (CCI; Supplementary Fig. 1D)[42]. Traumatic brain injury was shown to trigger microgliosis associated with an upregulation of DAM markers[43,44]. 72 h post injury, *Trem2*[wt/wt] as well as *Trem2*[KI/wt] mice showed an upregulation of total *Trem2* mRNA on the ipsilateral site compared to naïve mice (Supplementary Fig. 1E). As expected, in *Trem2*[KI/wt] mice we found an upregulation of both the *mKate2* reporter RNA (Supplementary Fig. 1F) and the unmodified wt *Trem2* RNA (Supplementary Fig. 1G). The DAM markers *Clec7a*, and *Cd68*[8] as well as *Grn* were increased to a similar extent in *Trem2*[wt/wt] and *Trem2*[KI/wt] mice (Supplementary Fig. 1H), suggesting that microglial transition to a DAM state is not altered by the genomic insertion of the reporter.

To examine the dynamics of TREM2 expression in the presence or absence of amyloid plaques, we crossed the reporter line with the *APP/PS1* mouse[45], a model of amyloidosis, to generate *Trem2-mKate2*[KI/wt].*APP/PS1*[tg/wt] (hereafter called disease) mice and compared them to *Trem2-mKate2*[KI/wt] (hereafter called healthy) mice (Fig. 1G). Histological analyses of 4-month-old mice revealed detectable mKate2 expression in microglia predominantly but not exclusively in close proximity to plaques (Fig. 1H). Calculating the percentage of mKate2 positive staining around plaques showed a gradual upregulation of the signal with increasing plaque proximity (Fig. 1I), suggesting the existence of microglia subpopulations with varying expression levels of TREM2.

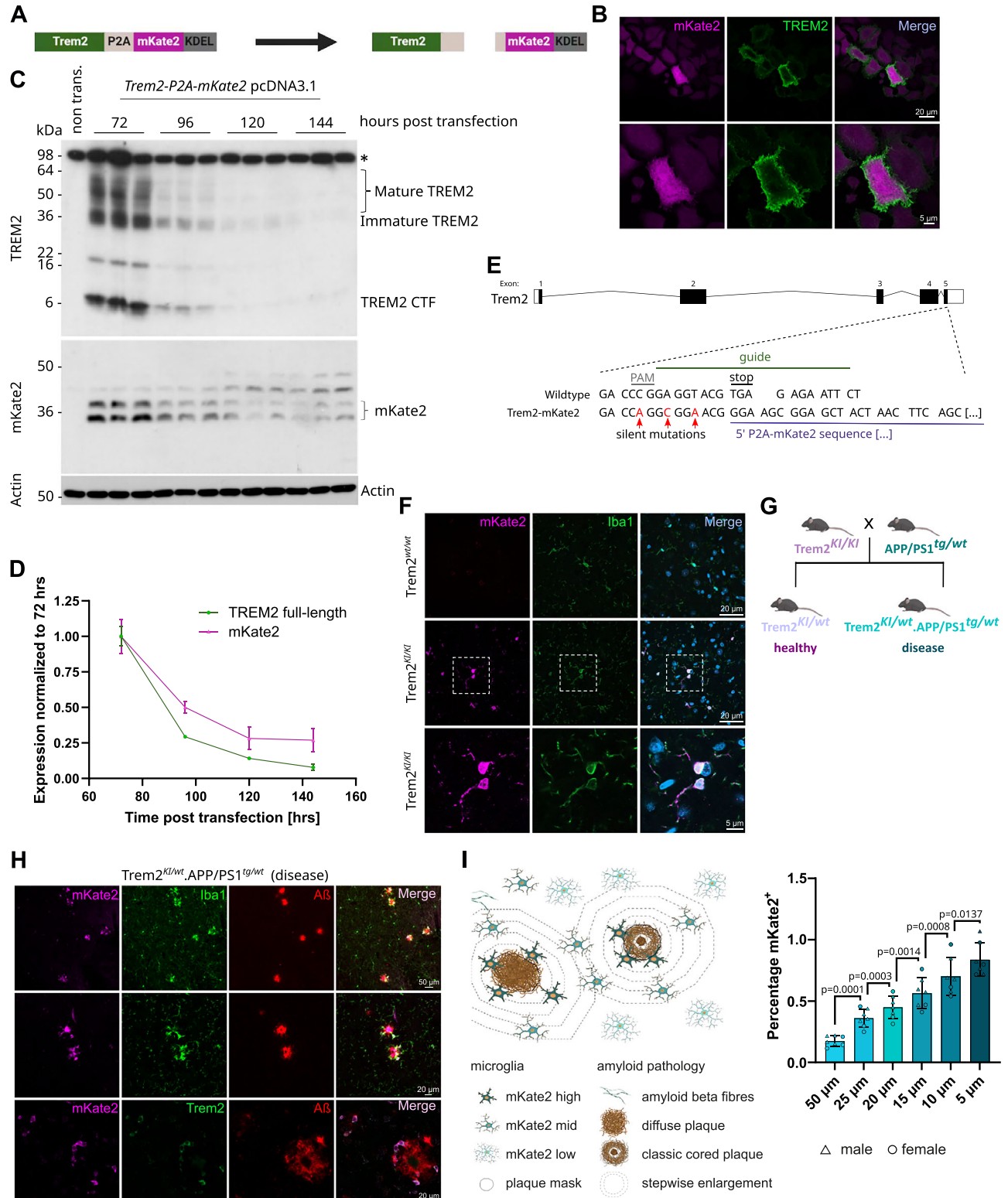

## TREM2 levels determine expression of functionally distinct microglial gene modules

To further characterize potential microglial subpopulations, we isolated total brain microglia from 9-month-old animals and performed fluorescence activated cell sorting (FACS) to separate microglia based on the mean fluorescence intensity of the mKate2 reporter as a proxy for TREM2 expression. We defined mKate2 low-, mid- and high expressing subpopulations in disease animals, and low- and mid mKate2 expressing microglial subpopulations in

healthy animals (Fig. 2A). As expected, high mKate2 expressing subpopulations were almost absent in healthy animals (Fig. 2A), although the relatively wide range of mKate2 expression observed in these animals suggests microglial subpopulations expressing varying amounts of TREM2 exist even under physiological conditions (Fig. 2A). This is consistent with an age dependent increase of the TSPO-PET signal and the formation of microglia nodules at sites of small myelin damage previously observed in aged wildtype mice[24].

**Fig. 1 | A TREM2 reporter mouse reveals gradual upregulation of TREM2 in microglia. A** Schematic of the reporter construct and the proteins expressed in vitro. Created in BioRender. Mühlhofer, M. (https://BioRender.com/mls61se. **B** Representative immunofluorescent images of HeLa cells transfected with the reporter construct shown in (**A**). mKate2 and TREM2 stainings are displayed in magenta and green. Scale bars 20 μm and 5 μm. **C** Immunoblot analysis of time-dependent degradation of TREM2 and mKate2 upon transient overexpression in HeLa cells. Various TREM2 species including the C-terminal fragment (CTF) are indicated. Shown are three technical replicates from a single experiment. Actin was used as a loading control. The asterisk indicates an unspecific signal. Non trans., non-transfected HeLa cells. Uncropped images in Source Data. **D** Quantification of the immunoblot data shown in (**C**). For TREM2, the immature and mature TREM2 signals were quantified while for mKate2 the two signals below and above the 36 kDa marker band were quantified. **E** CRISPR strategy to create the *Trem2-mKate2* reporter mice. Sites of silent mutations serving genotyping purposes are indicated. **F** Representative immunofluorescent images of cortical sections of *Trem2-mKate2* mice (wt and homozygous knock-in) showing positive labelling of mKate2 in knock-in animals only (magenta). Dashed squares represent the area of the zoom in in the bottom row. Iba1 and DAPI stainings are displayed in green and blue. Scale bars 20 μm and 5 μm. **G** Breeding scheme used to create the experimental mice for the *Trem2-mKate2* $^{Kl/wt}$.*APP/PS1* $^{tg/wt}$ strain. Only heterozygous reporter mice were used throughout the study. Non-transgenic littermates are referred to as 'healthy' and *APP/PS1* transgenic mice are referred to as 'disease'. **H** Representative immunofluorescent images of hippocampal sections of disease mice (4 months). mKate2 is displayed in magenta, Iba1-positive microglia or TREM2 in green and amyloid beta (Aβ) plaques in red. Scale bars 50 μm and 20 μm. **I** Percentage of mKate2+ signal with decreasing distance to a plaque. Shown is the mean ± SD. Male and female mice are indicated by triangles and circles. One data point represents averages of multiple images per mouse. Statistical analysis was carried out by repeated measures one-way ANOVA with Tukey's multiple comparisons test. N = 7 mice, age = 4 months. Created in BioRender. Mühlhofer, M. (https://BioRender.com/mls61se). Source data are provided as a Source Data file.

To characterize the biological relevance of microglial sub-populations stratified by TREM2 expression and disease pathology, we subjected sorted mKate2 low, mid, and high microglia to bulk RNA-seq (workflow Supplementary Fig. 2 **A**). Transcriptomic analysis confirmed that the microglial subpopulations identified via flow cytometry expressed different levels of *Trem2* as *Trem2* mRNA levels across microglial subpopulations showed a gradual increase (Fig. 2B). To explore whether microglia with different TREM2 levels were characterized by distinct transcriptional programmes, we performed PCA on the 5000 most variable genes (Fig. 2C). Assessment of low-dimensional embedding revealed distinct clusters for each group. PC1 explained almost 60% of the variance in the data and separated the clusters based on TREM2 levels, demonstrating a major impact of TREM2 expression on global transcriptomic responses. PC2 further split the clusters according to disease state (Fig. 2C).

Subsequently, we determined to what degree these TREM2-driven transcriptomic differences relate to amyloid plaque proximity by performing a gene set variation analysis (GSVA) of a plaque-induced gene (PIG) signature previously defined based on spatial transcriptomics and single-cell data (Fig. 2D)[46]. Overall, the PIG signature significantly distinguished the microglia subpopulations based on their health status and TREM2 level compared to other random signatures of the same size (Supplementary Fig. 2B). We observed an enrichment of the PIG signature only in microglia from mid and high mKate2 disease conditions with a gradually increasing enrichment score, while the PIG signature was not increased in the mid mKate2 healthy and low mKate2 disease subpopulation, whereas the low mKate2 healthy subpopulation was even negatively enriched (Fig. 2D). On the other hand, within the mid mKate2 healthy and low mKate2 disease microglia the PIGs ranking resembled the enrichment pattern of random signatures on population and sample level (Fig. 2D, Supplementary Fig. 2C). To assess the ranking of the PIGs on a population level, a gene set enrichment analysis (GSEA) was performed, corroborating positive enrichment of signature genes in mid and high mKate2 disease subpopulations and negative enrichment in the low mKate2 healthy subpopulation. (Supplementary Fig. 2D). In addition, enrichment of the PIG signature in single-nucleus RNA-seq data from human postmortem microglia of the dorsolateral prefrontal cortex and middle temporal gyrus[47] (Supplementary Fig. 2E) revealed that increased *TREM2* expression levels of *TREM2*-positive microglia were also associated with a higher inferred plaque proximity in humans with high AD neuropathological change (ADNC) significantly distinguishing them from donors with no ADNC (Supplementary Fig. 2F). To further validate the microglial subpopulations, we performed cell state deconvolution based on linear support vector regression (LSVR) utilizing reference single-cell RNA-seq data sets previously reported in two independent studies employing the 5xFAD mouse model

(Supplementary Fig. 2G homoeostatic vs DAM[8], Supplementary Fig. 2H methoxy XO4 positive vs negative[48]). Similarly, increases in DAM and methoxy XO4 signatures were measured almost exclusively in diseased animals in a TREM2-dependent manner. The comparisons with previously published microglial phenotypes (PIG, DAM, XO4 +) confirm that these signatures develop in response to Aβ pathology and demonstrate that these expression programmes increase with *Trem2* expression levels.

Our data indicated the presence of independent TREM2-driven as well as disease-driven transcriptional alterations. To delineate the disease-dependent gene expression programme, we compared microglia subpopulations with the same mKate2 levels and differing health status (low mKate2 disease microglia vs low mKate2 healthy microglia, mid mKate2 disease microglia vs mid mKate2 healthy) (Fig. 3A), and intersected the differentially expressed genes (DEGs) of both comparisons, which resulted in a signature of 106 DEGs (from here on defined as AD-related DEGs) (Supplementary Data 1). Heatmap visualization (including high disease) clearly shows the differences between healthy and disease for both groups even though the expression levels of the majority of these genes including the DAM-associated genes *Lpl*, *Spp1*, and *Csf1*[8] gradually increased with increasing TREM2 levels (Fig. 3B). Next, we defined genes directly related to TREM2 levels, by performing comparisons between groups with different TREM2 levels but matching health status (mid mKate2 healthy vs low mKate2 healthy, mid mKate2 disease vs low mKate2 disease, high mKate2 disease vs low mKate2 disease and high mKate2 disease vs mid mKate2 disease) (Fig. 3C). Intersection of those comparisons resulted in 65 DEGs (from here on defined as TREM2-related DEGs) such as *Cd36* and *Apod* involved in the Aβ and stress response of the aging and degenerative brain[49,50] (Fig. 3D, Supplementary Data 1). As expected, GSVAs of both the AD-related and TREM2-related DEGs significantly split the microglia according to their health status and TREM2 level, respectively, outperforming random signatures of the same size (Supplementary Fig. 3A, 3B). Moreover, the DAM-associated genes *Clec7a* and *Apoe*[8] were also identified in several of the comparisons (Fig. 3E). Like *Trem2* (Fig. 2B), both genes showed a gradual increase in expression in the disease condition from low to high mKate2; however, in the healthy condition a significant decrease in expression from low to mid mKate2 was observed, indicative of a differential transcriptional programme associated with TREM2 levels in healthy animals. The TREM2 downstream signalling genes *Hcst* (coding for Dap10 protein) and *Tyrobp* (coding for Dap12 protein), known to be involved in Syk-independent[19] and Syk-dependent TREM2 signalling[19,51], had a similar expression distribution compared to *Trem2* across conditions (Fig. 3F), while *Syk* itself, showed a decrease in expression from low mKate2 healthy to the high mKate2 disease condition, possibly indicating a negative feedback loop. Interestingly, *Mtor*, a major metabolism and energy sensor and regulator of

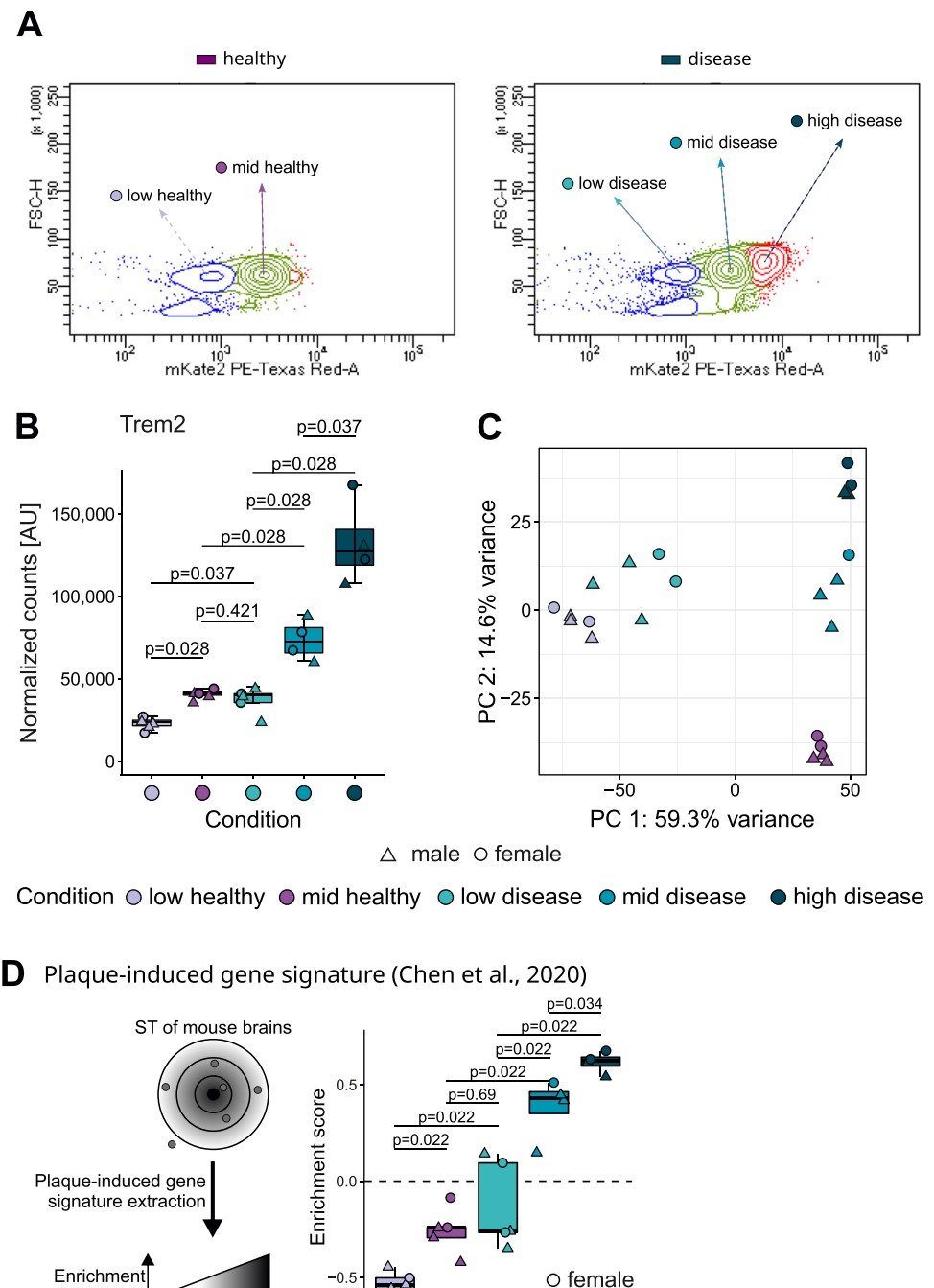

**Fig. 2 | Upregulation of microglial plaque-induced signature correlates with TREM2 expression level. A** Example contour plots showing the rationale for the FACS gating strategy in healthy and disease mice based on the amount of mKate2 positivity as a proxy for TREM2 expression. Subpopulations were sorted and named based on the amount of mKate2 present. **B** Boxplot of normalized *Trem2* expression coloured by condition. Boxplots show the 25%, 50% (median) and 75% percentile; whiskers denote 1.5 times the interquartile range (applies to all following boxplots). For the RNA-seq data of *Trem2-mKate2$^{KI/wt}$.APP/PS1$^{tg/wt}$* mice: N = 5 mice for low healthy, mid healthy, and low disease conditions; N = 4 mice for mid disease and high disease conditions; age = 9 months for all conditions. **C** PCA of the top 5000 most variable genes coloured by condition. **D** Boxplot of gene set variation analysis (GSVA) enriching for the plaque induced gene signature from Chen et al. coloured by condition. Male and female mice are indicated by triangles and circles. Statistics were calculated with an unpaired, two-sided Wilcoxon test followed by a Benjamini-Hochberg adjustment (**B**, **D**). Source data are provided as a Source Data file.

pathways downstream of TREM2, was equally expressed across all conditions (Fig. 3F)[19,27].

To dissect and compare additional transcriptional programmes of the stratified microglia subpopulations related to TREM2 function versus disease pathology, we performed a gene co-expression network analysis. This approach enabled the identification of co-regulated functional gene expression modules, providing a systems-level perspective that complements the DEG-based analysis and reveals broader regulatory patterns across conditions. The resulting network was defined by six major gene modules, containing 4002 genes. When

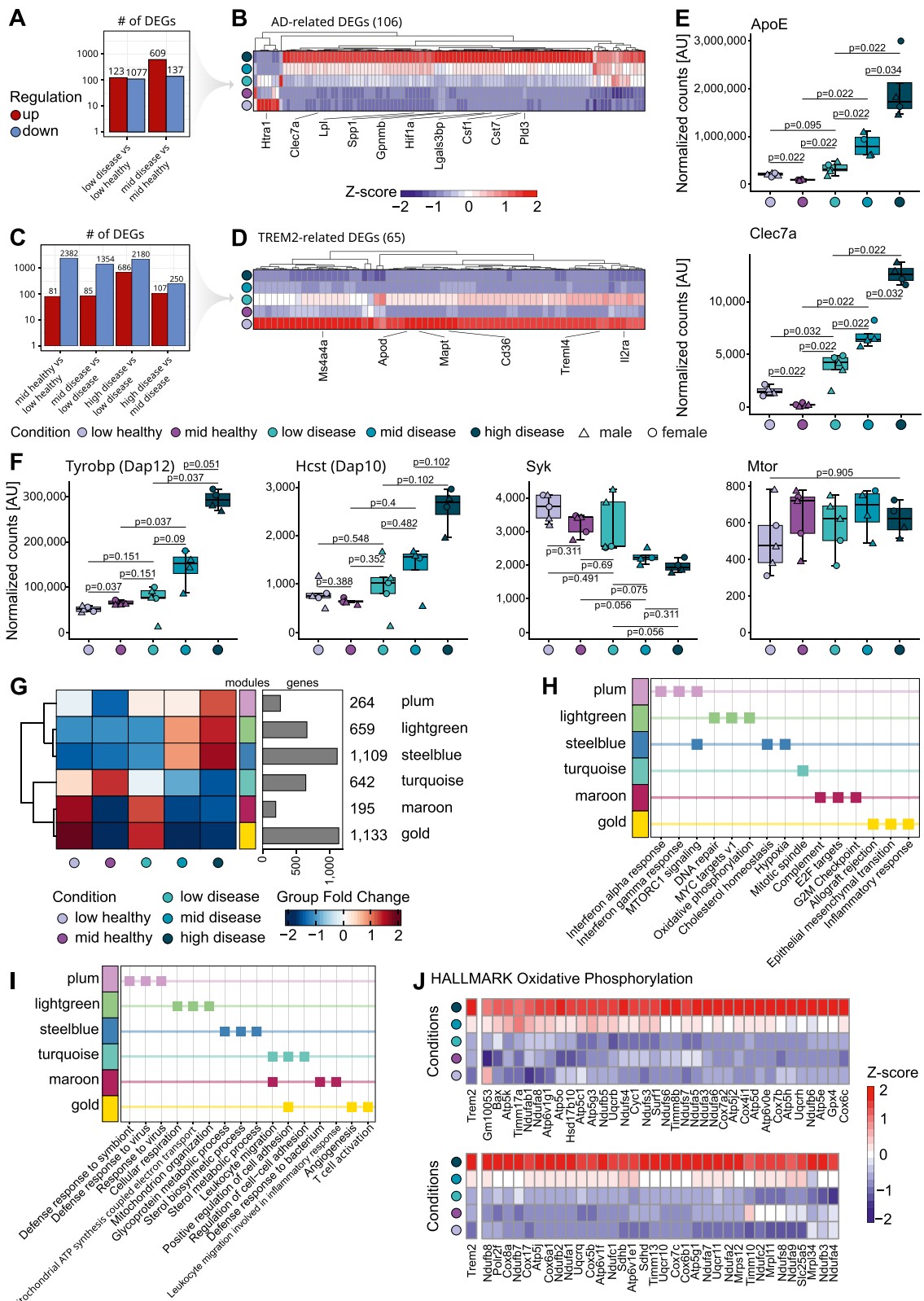

plotting condition-specific expression patterns in a heatmap, distinct differences were revealed (Fig. 3G). The plum, light green and steel blue modules showed an enrichment in the high mKate2 disease followed by the mid mKate2 disease conditions. Interestingly, the plum module was enriched for type I and II interferon response genes (*Ifitm3, Stat1, Irf7, Irf9*; Fig. 3H), which were recently identified to be uniquely upregulated after treatment with a TREM2 agonist[26]. Changes

in metabolic activity defined both the light green and steel blue module as indicated by an increased expression of genes involved in 'oxidative phosphorylation' (*AtpSj, Cox5b, Uqcrq*) in the former and 'sterol metabolic processes' (*Apoe, Cebpa*) and 'cholesterol homeostasis' (*Cd9, Lpl, Tmem97*) in the latter, which also comprised the majority of the AD-related DEGs (Figs. 3H, I, and Supplementary Fig. 3C). Further, a closer inspection of the genes involved in the

**Fig. 3 | Functionally distinct microglia gene modules are differentially driven by TREM2 expression level. A** Bar plot of DEGs between healthy and disease mice with matching TREM2 level coloured according to the mode of regulation (red: up; blue: down). **B** Clustered heatmap of AD-related DEGs from comparisons in A. The high disease condition was also plotted for comparison. Each DEG is coloured by its normalized, z-scored gene expression. Selected DEGs are highlighted. **C** Bar plot of DEGs between different TREM2 levels with matching health status coloured according to the mode of regulation. **D** Clustered heatmap of TREM2-related DEGs from comparisons in C. Each DEG is coloured by its normalized, z-scored gene expression. Selected DEGs are highlighted. **E** Boxplots of normalized gene expression of *ApoE* and *Clec7a* coloured by condition. Boxplots show the 25%, 50% (median) and 75% percentile; whiskers denote 1.5 times the interquartile range (applies to all following boxplots). **F** Boxplots of normalized gene expression of *Tyrobp* (coding for Dap12 protein), *Hcst* (coding for Dap10 protein), *Syk* and *Mtor* coloured by condition. **G** Clustered heatmap of gene co-expression network modules split by condition and coloured by group fold change (left), and bar plot of number of genes included in the respective module (right). **H** Overview of Molecular Signature Database (MSigDB) Hallmark geneset enrichment of module genes displaying the top 3 associated terms per gene module based on the Benjamini-Hochberg adjusted p-value. Squares are coloured by module. **I** Overview of GO enrichment of gene modules displaying the top 3 associated terms per gene module based on the adjusted p-value. Squares are coloured by module. **J** Heatmap of genes associated with the Hallmark term 'Oxidative phosphorylation' in the light green module coloured by normalized and z-scored expression and ordered by condition. Statistics were calculated with an unpaired, two-sided Wilcoxon test followed by a Benjamini-Hochberg adjustment (**E**, **F**). Male and female mice are indicated by triangles and circles. Source data are provided as a Source Data file.

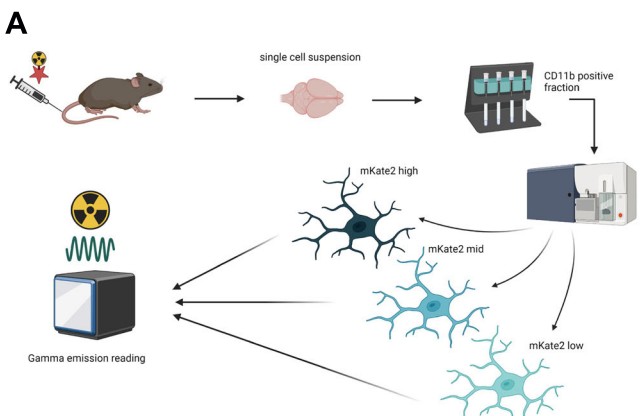

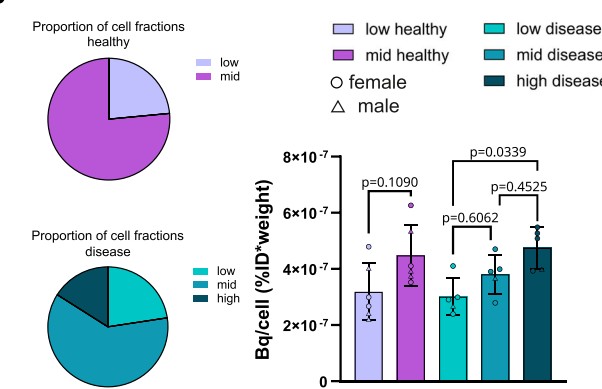

**Fig. 4 | Glucose uptake is associated with TREM2 upregulation. A** Schematic showing the workflow of scRadiotracing following radiolabelled [¹⁸F]-FDG injection in 14-months-old *Trem2-mKate2^{KI/wt}.APP/PS1^{tg/wt}* mice. Created in BioRender. Mühlhofer, M. (https://BioRender.com/mls61se. **B** Quantification of radiotracer uptake on a per cell basis in sorted microglial subpopulations after radiolabelled FDG injection. Left: proportion of subpopulations; right: quantification of FDG uptake on a per cell basis. N = 6 for healthy mice and N = 5 for disease mice. Shown is the mean ± SD. Statistical analysis was carried out by one-way ANOVA with Tukey's multiple comparisons test. Male and female mice are indicated by triangles and circles. Source data are provided as a Source Data file.

hallmark 'oxidative phosphorylation' revealed a gene expression pattern matching the *Trem2* expression level of the respective microglial condition (Fig. 3J). The gold module, including 60 out of 65 TREM2-related DEGs (Supplementary Fig. 3C), and maroon module consisted of genes with high expression levels in microglia with low mKate2 levels, independent of disease state (Fig. 3G) and were functionally related to genes involved in 'G2M Checkpoint' (*Mki67*), suggesting they have proliferation potential together with an enrichment of genes required for 'Inflammatory response' (*Cd40*, *Tnfaip6*) (Fig. 3H) and 'T-cell activation' (*Cd74*, *Clec4a1*) (Fig. 3I). Taken together, the transcriptome analysis of microglial subpopulations based on TREM2 expression levels in healthy and diseased mice revealed both TREM2-as well as disease-related gene modules, indicating differential transcriptional responses resulting from TREM2 expression which is further shaped by exposure to amyloid plaques.

### Gradual upregulation of TREM2 in plaque proximity is associated with increased glucose uptake

We previously observed that microglia alter their energy metabolism in the presence of a pathological challenge[52]. In line with this finding, we demonstrated that glucose uptake in microglia is TREM2-dependent and modulated by activation state[25]. Here, our bulk RNA-seq data revealed that genes involved in oxidative phosphorylation were altered in the subpopulations with different amounts of *Trem2* mRNA levels. To directly examine if and how TREM2 levels in microglial subpopulations could influence energy metabolism, we injected 14-month-old mice with 45 ± 3 MBq [¹⁸F]FDG. CD11b-enriched microglia were sorted into low, mid, and high expressing subpopulations as defined by mKate2 levels and radioactivity was measured (Fig. 4A)[53]. The proportion of mKate2 low microglia was roughly 25% in both healthy and disease mice (Fig. 4B) while the mKate2 high subpopulation was largely absent in healthy mice. Glucose uptake increased with mKate2 expression although it reached statistical significance only in disease mice (Fig. 4B). The unexpectedly similar levels of glucose uptake by mid mKate2 healthy and mid/high mKate2 disease microglia may be due to the downregulation of Syk signalling in the mid/high mKate2 disease microglia (see Fig. 3F) or indicate that the amount of TREM2 expressed by mid mKate2 healthy microglia might be sufficient to trigger maximal glucose uptake.

### Distinct metabolic profiles are associated with TREM2 expression levels

Since the above findings suggest that increased glucose uptake and gene expression related to metabolic changes correlate with increased TREM2 expression in healthy and diseased mice, we sought to investigate and validate whether TREM2 regulates the microglial metabolic state in the presence and absence of amyloid pathology. Metabolomic and lipidomic analyses were conducted in sorted healthy and disease microglial subpopulations to identify major trends in metabolic processes according to diseases status or TREM2 expression levels. Then, principal component analysis (PCA) was performed to reduce the dimensionality of the data. Consistent with glucose uptake, liquid chromatography-mass spectrometry (LC-MS) findings suggest that TREM2 levels, rather than amyloid pathology, drive component

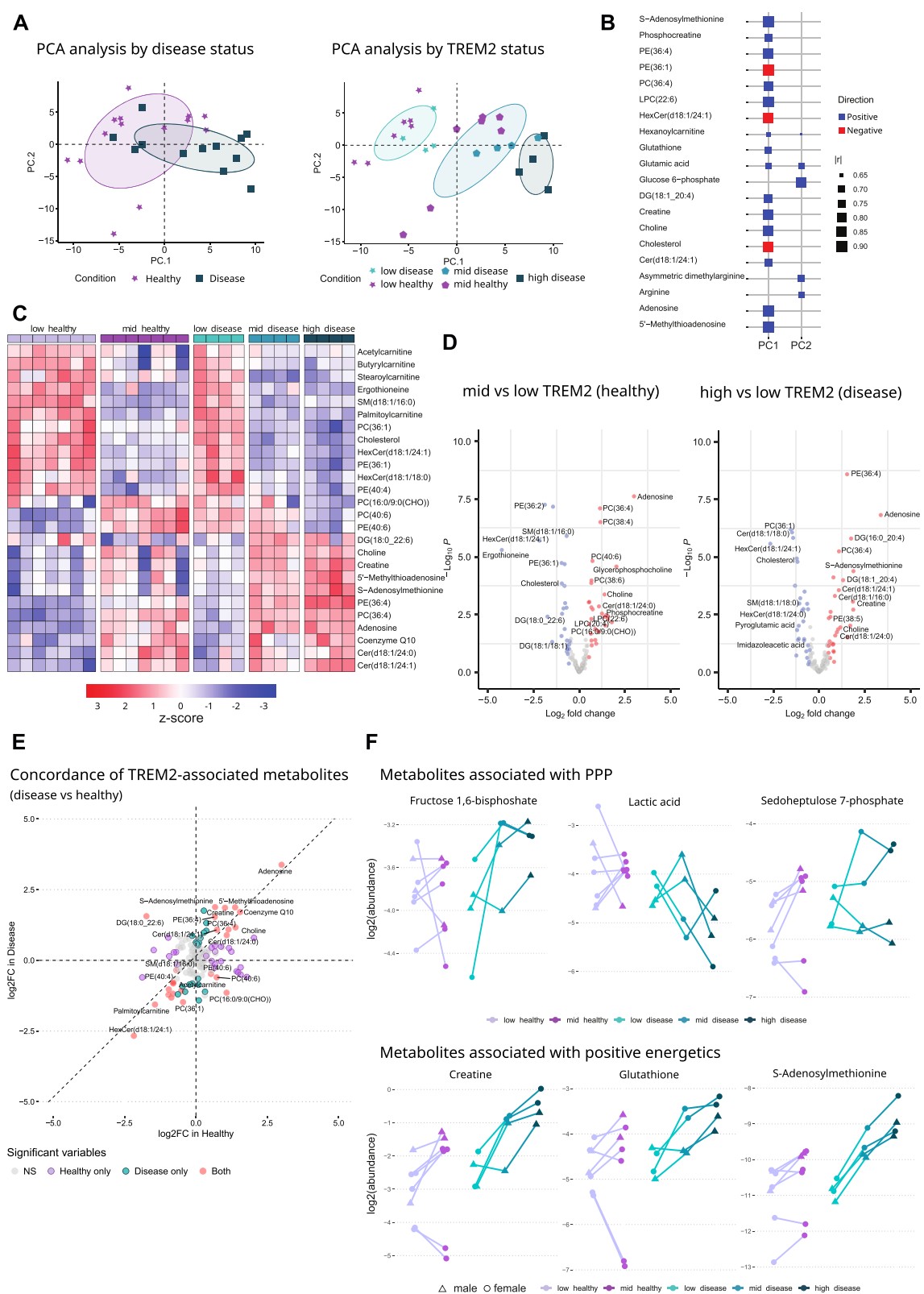

stratification (Fig. 5A). As such, TREM2 expression tracked more closely with patterns of metabolic variations along the PC1 axis than disease status (Fig. 5B). PC1 scores were positively contributed by cellular energetic substrates, such as adenosine, S-adenosylmethionine (SAM), phosphocreatine, and were negatively correlated with cholesterol and glycosphingolipid hexosylceramide (HexCer (d18:1/24:1); Fig. 5B). The most significantly changed analytes in the comparison of high vs. low

mKate2 disease displayed on the heatmap (Fig. 5C) included many of the energetic metabolites identified in the PC contributor analysis. These metabolites were significantly changed with consistent directionality driven by TREM2 levels in both healthy and disease mice shown in volcano plots (Fig. 5D, Supplementary Data 2). Consistent with this pattern, free cholesterol ($-1.2$ $\log_2$ FC) and palmitoylcarnitine ($-1.6$ $\log_2$ FC) levels were both significantly ($p < 0.0005$) lower in high

**Fig. 5 | Microglial TREM2 levels correlate with cellular energetics metabolites and PPP intermediates. A** PCA analysis of 141 analytes measured in microglia sorted by mKate2 levels in healthy and disease mice from the *Trem2-mKate2 $^{KI/wt}$.APP/PS1 $^{tg/wt}$* line. N = 7 and N = 4 for healthy and disease mice, respectively; age = 9 months. **B** Top 20 highest contributors to PC1 and PC2 (lrl: correlation coefficient). **C** Heatmap displaying most significantly changed lipids/metabolites within the comparison of high vs. low mKate2 (i.e., TREM2) in disease mice. **D** Volcano plots showing significantly changed ($p < 0.05$ and log2(fold change) > 0.5) lipids/metabolites in sorted microglia by mKate2 levels in healthy and disease mice. For each metabolite, an unpaired, two-sided moderated t-test was performed to assess differential abundance between experimental groups. *P* values were adjusted for multiple comparisons using the Benjamini-Hochberg FDR procedure. **E** Concordance analysis of metabolites associated with TREM2 level differences in healthy vs. disease states. Significantly ($p < 0.05$ and log2(fold change) > 0.5) changed analytes are highlighted according to disease status, and those modified in both healthy and disease mice are labelled in red. **F** Abundances of metabolites associated with PPP and positive energetics vs. mKate2 (i.e., TREM2) levels plotted on a per mouse basis. N = 7 and N = 4 for healthy and disease mice, respectively. Male and female mice are indicated by triangles and circles. Source data are provided as a Source Data file.

vs. low TREM2 microglia, suggesting increased lipid catabolism and cholesterol efflux. To interrogate the concordance of the metabolites with TREM2 expression and disease status, we compared fold-changes in disease vs. healthy mice (Fig. 5E). Many metabolites associated with positive cellular energetics such as S-adenosylmethionine, coenzyme Q, and creatine were increased in subpopulations expressing higher TREM2, in both healthy and disease contexts. Visualization of the data along the axis of TREM2 expression per mouse shows that increasing TREM2 correlates with higher abundance of fructose-1,6-bisphosphate, sedoheptulose-7-phosphate, creatine, glutathione (GSH) and SAM, and reduced abundance of lactate. Taken together, this data suggests increased glucose uptake correlates with higher pentose phosphate pathway (PPP)[54], methylation and redox capacity in high disease or mid healthy vs. low TREM2 expressing microglia (Fig. 5F). Overall, increased TREM2 levels are associated with elevated energetic and metabolic capacity, and improved cholesterol handling in microglia.

## Gradual increase of microglial metabolic capacity correlates with enhanced phagocytic capacity

Given that expression of genes associated with metabolic activity (Figs. 2 and 3), extent of glucose uptake (Fig. 4) and levels of key metabolites associated with positive cellular energetics (Fig. 5) were all gradually increased in the mid and high mKate2 subpopulations, we next asked whether this would translate into changes of microglial function. To test this hypothesis, we performed an ex vivo phagocytosis assay where microglia from *Trem2-mKate2 $^{KI/wt}$.APP/PS1 $^{tg/wt}$* animals (aged 9-10 months) were incubated with fluorescent pHrodo-labelled myelin and subsequently subjected to FACS to determine the amount of myelin taken up by each of the subpopulations. We first defined low, mid, and high mKate2 gates in the CD11b-positive cell population (Fig. 6A, B). As a next step, we defined pHrodo gates for the low, mid and high mKate2 gates to quantify the percentage of pHrodo-positive cells (Fig. 6C). This revealed significantly higher phagocytic capacity of high TREM2 microglia compared to low and mid TREM2 microglia (Fig. 6D). Quantification of median fluorescence intensity (MFI) demonstrated a gradual increase in phagocytic capacity across microglial subpopulations (Fig. 6E). As expected, microglia incubated at 4 °C did not show any uptake of fluorescently labelled myelin (Supplementary Fig. 4A). To assess whether gradually increased phagocytic capacity in microglia with increasing TREM2 expression is a consistent feature across independent mouse models of amyloid pathology we next conducted the same experiment with microglia from *App* $^{SAA}$ knock-in mice[52] crossed to our TREM2 reporter mice. We followed the same gating strategy as used for *Trem2-mKate2 $^{KI/wt}$.APP/PS1 $^{tg/wt}$* animals (Supplementary Fig. 4B). Consistent with our findings in *APP/PS1* mice, microglia derived from *App* $^{SAA}$ mice showed a similar gradual increase in phagocytic capacity across microglial subpopulations (Fig. 6F, 6G). Uptake of fluorescently labelled myelin was again not detected when incubating microglia at 4 °C (Supplementary Fig. 4C). These findings demonstrate that the mKate2 reporter allows to monitor distinct functional states of microglia in the context of amyloid pathology in two independent mouse models.

## A TREM2 agonist antibody induces metabolic changes dependent upon TREM2 expression level

We previously demonstrated that a mouse TREM2 agonist antibody (4D9)[36] engineered to cross the blood brain barrier using an antibody transport vehicle (ATV:4D9) boosts TREM2 activity in vivo and increases microglial metabolism[26]. To determine the effect of this brain-penetrant antibody on microglial subpopulations stratified by TREM2 expression levels, mice were dosed monthly with 1 mg/kg ATV:4D9 or isotype control for 4 months (Fig. 7A, and Supplementary Fig 5A, B). Here, *Trem2-mKate2 $^{KI/wt}$.App $^{SAA/SAA}$.hTfR $^{KI/KI}$* mice were used to enable the antibody to cross the blood brain barrier by binding to human TfR[26,55], as well as to validate our results in a second amyloid model. Mice received the last antibody dose 1 month prior take down. In addition, 30 mg/kg of fluorodeoxyglucose (FDG) was dosed 1 h before animal takedown to enable detection of phospho-FDG (pFDG) in sorted microglia (see Supplementary Fig. 5C for validation of the method). After perfusion, brains were dissociated and microglia were sorted by flow cytometry into low, mid, and high TREM2 expressing subpopulations (Supplementary Fig. 5B). Cell lysates were then subjected to LC-MS (Fig. 7A). PCA analysis and loading scores (Figs. 7B, C) of the LC-MS data show that the low TREM2 subpopulations derived from mice dosed chronically with ATV:4D9 and isotype control have similar lipidomic and metabolomic profiles, while the mid and high TREM2 subpopulations are distinct and cluster depending upon antibody dosing. Interestingly, high TREM2 isotype and mid TREM2 ATV:4D9 cluster together. Likewise, mid TREM2 isotype and high TREM2 ATV:4D9 form one cluster (Fig. 7B), suggesting that the TREM2 agonist antibody differentially modulates microglial metabolism dependent on TREM2 expression level. To verify that these changes were not caused by changes to the microglia activation state, sorted subpopulations from the same animals were additionally subjected to bulk RNA-seq. Assessment of the expression of key DAM genes such as *Trem2, Apoe, Clec7a* and *Tyrobp* in the bulk RNA-seq data revealed a gradual increase in low, mid, and high TREM2 microglia, similar to what we observed in *APP/PS1* mice (Supplementary Fig. 5D, **compare** Fig. 2B **and** Figs. 3E, F). However, we did not see any significant changes between ATV:4D9 and isotype control groups. Given that mice were collected one month after the final antibody dose it was to be expected that the antibody would have little effect on the transcriptome[26]. In agreement with this, mass spectrometry based proteomics did not detect any differences in TREM2 levels neither in whole brain RIPA extracts (Supplementary Fig. 5E, F) nor in CSF (Supplementary Fig. 5G, H). Further analysis of the LC-MS based lipid and metabolite data revealed the component separation is driven by phospholipids such as PCs and PEs, as well as glycolysis intermediate pFDG and lysosomal BMP lipid species. Consistent with PCA analysis, the heatmap (Fig. 7D) and volcano plots (Fig. 7E, and Supplementary Data 3) suggest that upon chronic dosing of ATV:4D9, very few significant changes are induced in the low TREM2 subpopulation, and the metabolic changes induced by ATV:4D9 are predominantly observed in the mid and high TREM2 expressing subpopulations (Figs. 7D, 7E, Supplementary Data 3). In contrast to the low TREM2 expressing microglia, mid TREM2 levels were associated with significant increases in key metabolic

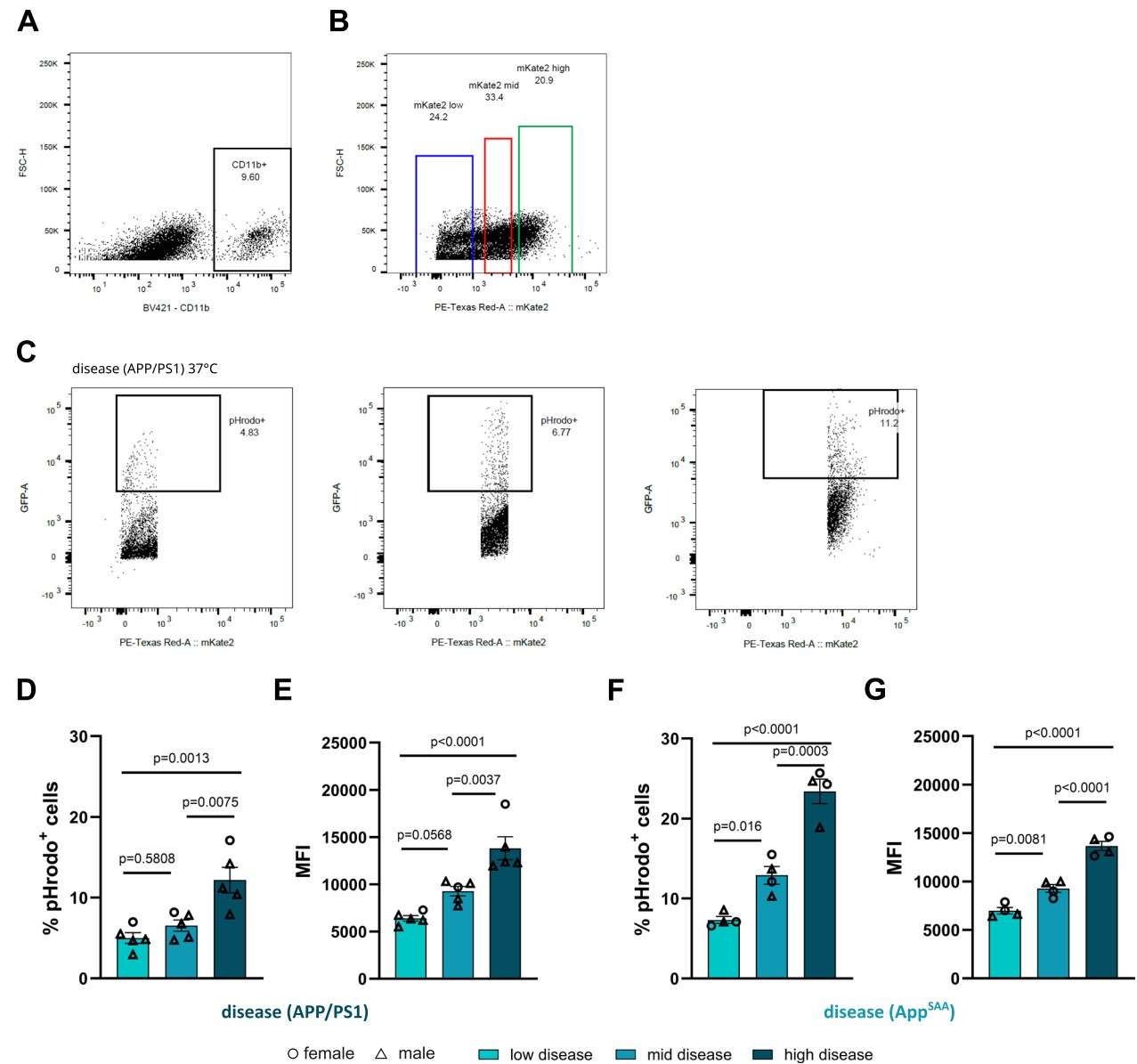

**Fig. 6 | Gradual increase in microglial metabolic capacity correlates with gradual increase in phagocytic capacity. A** Gating on CD11b-positive cells in single cell suspensions of whole brain. **B** Defining mKate2 low, mKate2 mid and mKate2 high gates in the CD11b-positive cell population. **C** Representative FACS data from a *Trem2-mKate2 $^{KI/wt}$.APP/PS1 $^{tg/wt}$* mouse showing the pHrodo gates in the mKate2 low, mid, and high subpopulations from which percentages of pHrodo-positive cells were quantified. **D**, **E** Quantification of percentage of pHrodo-positive cells (**D**) and median fluorescence intensity (**E**) in the mKate2 low, mid, and high subpopulations in 9-10-months-old *Trem2-mKate2 $^{KI/wt}$.APP/PS1 $^{tg/wt}$* mice (N = 5). **F**, **G** Quantification of percentage of pHrodo-positive cells (F) and median fluorescence intensity (G) in the mKate2 low, mid, and high subpopulations in 13-months-old *Trem2-mKate2 $^{KI/wt}$.APP $^{SAA/SAA}$.hTfR $^{KI/KI}$* mice (N = 4). Male and female mice are indicated by triangles and circles. Shown is the mean ± SEM. Statistical analyses in (**D**–**G**) were carried out by one-way ANOVA with Tukey's multiple comparisons test. Source data are provided as a Source Data file.

indicators of glycolysis (pFDG), lysosomal function (BMP(22:6/22:6)), and peroxisomal function (PE-(P16:0/20:4)) (Fig. 7**F**). Furthermore, ATV:4D9 treatment was highly effective in increasing levels of poly-unsaturated fatty acid (PUFA)-containing phospholipids, including sub-species of phosphatidylcholine (PC), phosphatidylethanolamine (PE), and plasmalogen phosphatidylethanolamine (PE(P)) in the mid-TREM2 subpopulation (Fig. 7F, and Supplementary Fig. 5I). Interestingly, while in the mid-TREM2 expressing microglia ATV:4D9 mediated increased levels of lipids and metabolites, the trend was reversed in the high-TREM2 expressing microglia (Fig. 7D–F), supporting the above PCA analysis (Fig. 7B). These findings indicate that the level of TREM2 expression on the cell surface of microglia is critical for mediating the metabolic response to the agonist antibody.

## Discussion

Our study further supports the idea that microglia can readily adjust their energy metabolism to resolve challenges[52]. We have shown previously that microglia have a high rate of glucose uptake and that this is dependent on their activation status[25]. Using single cell radiotracing, we demonstrate that microglial glucose uptake is strongly correlated with the expression level of TREM2. By combining glucose uptake studies with metabolomics, we also ascertain the likely fate of glucose in supporting the cellular redox environment required for glutathione and lipid synthesis. Furthermore, the gradual increase in glucose uptake and metabolic capacity with increasing TREM2 expression is concordant with the gradual increase in phagocytic capacity suggesting the crucial role TREM2 plays in regulating metabolic activity likely

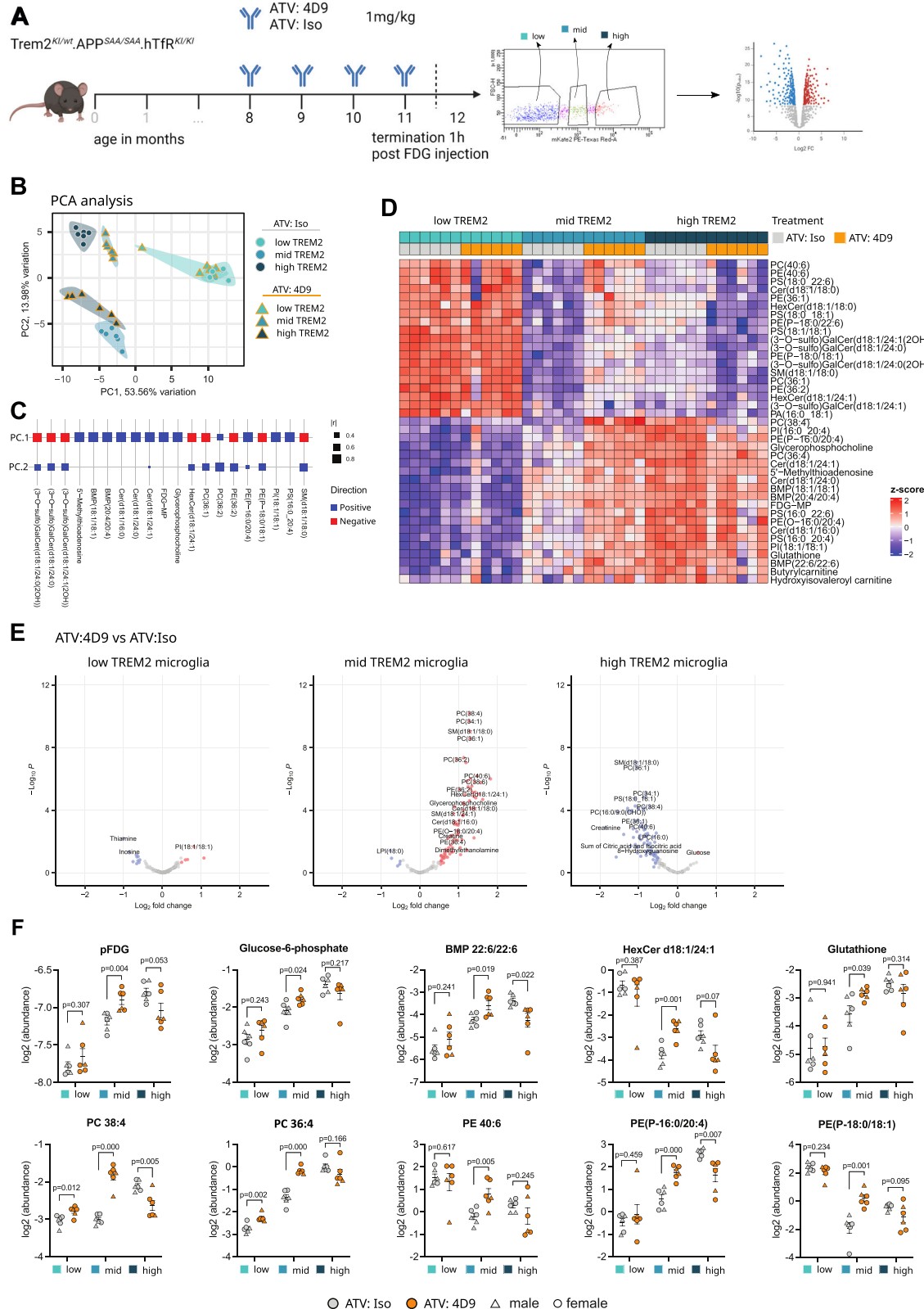

supports functions such as microglial phagocytosis[15,20,56]. Microglia express the hexokinase 2 (HK2) isoform which preferentially diverts glucose for NADPH generation required for glutathione redox and lipid biosynthesis via the pentose monophosphate pathway (PPP)[57,58]. Although NADPH was not directly measured due to technical challenges, this interpretation is supported by our observations that 1) increased PPP intermediates occur at the expense of lower lactate, 2)

higher abundance of several unsaturated long chain PE and PC species, 3) increased total glutathione levels and 4) TREM2-associated increase in transcripts of lipid biosynthesis enzymes. In addition, cellular S-adenosylmethionine which is required for PC synthesis[59] was significantly upregulated in high vs low TREM2 expressing microglial subpopulations. Increased glutathione is inversely associated with pro-inflammatory potential by lowering ROS production[60], supporting

**Fig. 7 | TREM2 agonist ATV:4D9 modulates metabolic profile dependent upon TREM2 expression level. A** Schematic of experimental design. Created in BioRender. Mühlhofer, M. (https://BioRender.com/mls61se). 8-month-old *Trem2-mKate2*[KI/wt].*APP*[SAA/SAA].*hTfR*[KI/KI] mice were injected with 1 mg/kg ATV antibody monthly until 11 months. At 12 months, mice were injected with 30 mg/kg FDG, perfused 1 h later and then the single cell fractions of isolated brains were sorted into low, mid and high mKate2 expressing subpopulations. Cells were pelleted and frozen for further analysis. N = 6 mice per treatment group. **B** PCA analysis of 149 analytes measured by LC-MS in sorted microglia. **C** Top 20 highest contributors to PC1 and PC2 (|r|: correlation coefficient). **D** Heatmap of top 40 differentially regulated lipids and metabolites. **E** Volcano plots of treatment comparisons within the same mKate2/TREM2 level. For each metabolite, an unpaired, two-sided moderated t-test was performed to assess differential abundance between experimental groups. P-values were adjusted for multiple comparisons using the Benjamini-Hochberg FDR procedure. **F** Feature plots of individual lipids/metabolites altered by ATV:4D9 (orange symbols) vs ATV:Iso (grey symbols) within low, mid and high mKate2 subpopulations (N = 6 mice per treatment group). Shown is the mean ± SEM. *p* values were calculated within each group by unpaired Student's t-test (two-tailed). Male and female mice are indicated by triangles and circles. Source data are provided as a Source Data file.

cellular proliferation[61], and decreasing ferroptosis potential[62] in microglia. Furthermore, treating mice with the TREM2 agonist ATV:4D9 resulted in metabolic alterations akin to those observed along the axis of increasing endogenous TREM2 levels particularly in mid-expressing microglia. Interestingly, in high TREM2 expressing microglia ATV:4D9 resulted in attenuated metabolic profiles like those observed in mid-TREM2 expressing microglia with isotype treatment. These results demonstrate that TREM2 agonism can regulate microglial metabolic activity, while tempering activity in high-TREM2 expressing microglia. This suggests that TREM2 levels are a key regulator of metabolic state and further demonstrate that TREM2 has an immunomodulatory role whereby it can tune metabolic activity up or down depending upon a threshold of the TREM2 receptor. Increased accumulation of PUFA, PC, PE and plasmalogens suggests that ATV:4D9 could improve cell membrane flexibility that is required to support higher phagocytic and lipid clearance activities. Notably, plasmalogens, an abundant class of phospholipids, have been found to be deficient in AD patient brains, and some studies have supported beneficial effects of reduced inflammation and neuroprotection with plasmalogen administration[63–65].

Taken together, TREM2-associated increases in glucose uptake and its usage to support glutathione redox may be a key underlying mechanism by which TREM2 promotes microglial survival in times of stress. Our lipidomic and metabolomic data strongly suggest that TREM2 expression is a driver of lipid catabolic pathways, including fatty acid oxidation (FAO), which is consistent with the previously reported ability of TREM2 agonist antibodies to clear lipid droplets and promote FAO in iPSC-derived human microglia[26]. Importantly, largely based on KO studies, TREM2 has been identified as a major regulator of microglial cholesterol transport and metabolism[30,31,66]. Our study lends strong support to this notion by showing lower free cholesterol levels in TREM2 high microglia, consistent with higher catabolism or efflux capacity. Since cholesterol storage forms, like cholesteryl esters, are known to accumulate in AD brain[67], the impact of the TREM2 pathway on cholesterol clearance is likely to be disease relevant. Since TREM2 risk variants are thought to be loss-of-function, TREM2-driven glucose uptake is likely perturbed in individuals carrying these variants, which is supported by previous studies of the T66M mutation in mice[24], and is also in line with reduced cerebral glucose metabolism and regional hypometabolism reported in a NHD patient[68]. There is a high prevalence of AD and metabolic disease being co-morbidities, as well as risk factors for each other[69–71], hinting towards a potential contribution of microglia metabolism in disease. Moreover, glucose hypometabolism may be a key manifestation of prodromal AD and may therefore be useful as a biomarker. However, direct comparison of these results should be conducted with caution, since macrophages in the periphery and microglia in the brain can be functionally diverse given the different environments. Surprisingly, we only detected minor differences in disease vs. healthy conditions, while the TREM2 level effects were largely concordant between the conditions. These findings indicate that regardless of brain environment, TREM2 can function as a metabolic driver to increase cellular bioenergetic capacity, likely in accordance with increased demands for innate immune functions.

Subjecting the TREM2 stratified subpopulations to bulk RNA-seq, we validated that sorting based on mKate2 levels is indeed a proxy for TREM2 expression differences. Using the published PIG signature by Chen et al.[46], we performed a gene set variation analysis assessing the data in a spatial context. In contrast to Chen et al.[46], isolated mouse microglia displayed a significant upregulation of the PIG signature with increasing TREM2 levels. This effect is enhanced in the disease context due to plaque presence in both mice and humans. Additionally, we observed a decreased enrichment score comparing mid healthy to low healthy mouse microglia and a milder increase in microglia of human donors with no ADNC[47]. This suggests that the PIG signature reflects a general microglial response to an environment requiring higher phagocytic capacity and not necessarily an amyloid-specific microglial response. While plaque vicinity constitutes an environment in which the signature has reached its maximum, it is dependent upon TREM2 upregulation. Hence, we postulate that the PIG signature reflects a cellular response enabling subpopulations of microglia to perform activities such as removing myelin to maintain brain homoeostasis[72]. Further comparing the gene signatures of the subpopulations to homoeostatic, stage 1 DAM, and stage 2 DAM as defined by Keren-Shaul et al.[8], we found that the majority of microglia in the healthy mice expressed a homoeostatic signature together with the minor presence of a DAM 1 signature which can be potentially explained by aging-related myelin damage that microglia clean-up[24,73,74]. Strikingly, an increasing fraction of the DAMs, specifically of stage 2 DAMs, was observed in disease microglia. This finding aligns well with the enrichment of the PIG signature in the disease context pointing towards higher plaque proximity required for the emergence of DAMs[8] dependent on the *Trem2* level but also suggests that the expansion of the *Trem2*-dependent stage 2 DAM population could be additionally regulated by the *Trem2* level. Additionally, we validated the reporter-based sorting approach by comparing the signatures to the classifications from Grubman et al.[48]. Here, we saw a highly consistent pattern in which disease conditions showed the expected upregulation of the 5xFAD methoxy-X04 positive signature with a concomitant WT signature reduction. To gain further insights into the functional state of the microglia with different TREM2 levels in healthy and disease conditions, gene co-expression network analysis was performed identifying six gene modules. One of the modules, plum, contained type I and II IFN response genes, which were specifically elevated in the disease context and in particular in high TREM2 microglia. A previous study recently identified a subset of DAM expressing increased levels of type I IFNs promoting cellular states detrimental to memory and cognition[75], but the upregulation of type I IFN in disease microglia based on plaque proximity and *Trem2*-level had not yet been documented[75]. Additionally, type II IFN is known to prime microglia for the induction of enhanced secretion of inflammatory molecules such as IL-6 and TNFα, giving rise to neurotoxic phenotypes that drive neurodegeneration[76]. Future studies are needed to understand the role that TREM2 levels and type I and II IFN signalling play in disease progression including studies addressing spatial distribution of this microglial subpopulation.

Proximity to amyloid plaques, an increased expression of stage 2 DAM genes, and/or the high expression of type I and II IFNs can cause a

high energetic demand on microglia. Supporting our lipidomic analysis, we observed an upregulation of genes associated with oxidative phosphorylation and lipid metabolism (e.g., the light green and steel blue modules, Fig. 3G-I) that correlated with *Trem2* mRNA expression levels. Overall, these findings extend the current understanding of TREM2-dependent regulation of microglial state and metabolism with higher resolution regarding TREM2 expression levels. We show that physiological TREM2 expression and function occurs as a gradient as opposed to an on/off switch.

By treating *Trem2-mKate2*[KI/wt].*App*[SAA/SAA].*hTfR*[KI/KI] mice chronically with a transport vehicle engineered TREM2 agonistic antibody 4D9, i.e., ATV:4D9[26,36] we demonstrate that microglia with intermediate expression levels of TREM2 responded the most, with increasing metabolites related to glucose uptake as well as antioxidants, reflective of an overall increased metabolic fitness. Some metabolites decreased in high vs mid subpopulations upon chronic dosing, suggestive of a ceiling effect. Similarly, too little expression of TREM2 may not provide enough target for the TREM2 agonist to be active, further demonstrating the importance of TREM2 expression levels on microglia[77]. These findings may have important consequences for the design of clinical trials examining TREM2 agonists. Based on our findings, careful consideration of TREM2 expression levels in patients could be an important aspect for determining the optimal time point for treatment. Given the recently documented slower cognitive decline in AD patients treated with anti-amyloid antibodies such as Lecanemab[78] or Donanemab[79] it is important to understand whether a combination therapy (i.e., co-dosing with an agonistic anti-TREM2 antibody) would improve efficacy. While we have recently shown that a single injection of an agonistic TREM2 antibody significantly increases brain-wide glucose metabolism on a timescale of days[26] the current study highlights TREM2 expression at the cellular level will be decisive in the exact metabolic outcome. Interestingly, treatment studies in mice[80] and humans[81] show that anti-amyloid antibody-mediated removal of amyloid plaques leads to a reduction in brain-wide glucose metabolism[80] and, consistently, lower expression of genes related to glycolysis[80,81], potentially due to a reduction in microglial activation as a direct consequence of plaque removal. It will thus be important to investigate the changes in microglial metabolism in a co-treatment paradigm where the exact outcome will depend on factors such as disease stage (animal model and age), dosing regimen (sequential vs. simultaneous) and antibody dose and duration where microglial exhaustion could impair efficacy of anti-amyloids over time[82].

Regarding ATV:4D9 dosing in this study it is important to note that while we did not detect any significant differences on transcriptomic level, we found profound alterations on the metabolomic level. This underscores the need to perform subpopulation-specific proteomic analyses to gain insight into the underlying molecular cause. Moreover, it will be crucial to further investigate the functional consequences of antibody-mediated metabolic changes in mid and high TREM2 microglia. As another limitation of our study, we note that the mKate2 reporter likely does not capture the entirety of microglia state heterogeneity. Indeed, it has been reported that microglia characterized by high TREM2 expression levels may be composed of subpopulations which are functionally different[83]. Our TREM2 reporter model would not allow to distinguish between such subpopulations.

Taken together, our data highlights the correlation of TREM2 expression with microglial transcriptome, metabolic activity and phagocytic capacity. Further, we provide evidence that transcriptional programmes linked to differential levels of TREM2 are further shaped by the disease state, here exemplified in an amyloid plaque disease model. Importantly, we found that TREM2 surface expression is the major driver for microglial metabolic profiles, which could be modulated further by a TREM2 agonistic antibody. Given that we observed TREM2 upregulation to be gradual in plaque proximity as well as the bidirectional modulation by the TREM2 agonistic antibody, our data

demonstrate the need to determine the optimal timepoint for therapeutic intervention for this approach in patients, which may have been missed in the failed INVOKE-2 study[37]. Non-invasive longitudinal monitoring of TREM2 expression by PET imaging is an emerging powerful approach that will help optimize timing during therapeutic intervention[84,85]. Finally, our TREM2 reporter mouse model allows for the functional characterization of distinct states of microglia which is crucial for understanding the implications of heterogeneity of this crucial cell type's role in brain health and disease.

## Methods

### Expression vector design and expression
To create the reporter mouse, we developed a plasmid-based construct. mKate2 was chosen as a fluorescent tag[38]. To ensure that the tag does not interfere with normal TREM2 expression, function or localization we inserted the P2A peptide sequence, which allows the separate expression of both proteins[39], between the C- and N-termini of TREM2 and mKate2, respectively. To avoid secretion of the reporter, we included a KDEL sequence after the mKate2 sequence to retain it in the endoplasmic reticulum (ER)[40]. To create the expression vector for the TREM2-P2A-mKate2-KDEL construct, we obtained genomic mouse DNA from C57Bl6/J animals from which the TREM2 sequence was amplified. We obtained the mKate2 plasmid from Addgene and amplified the sequence by standard polymerase chain reaction (PCR). Primers for amplification, which were designed with overhangs compatible with the Gibson assembly cloning, are as follows: ctggc tagcgtttaaacttaGGCCAAGCTTGCCACCATG (Trem2-fwd), ctccag cctgCTTCAGCAGGCTGAAGTTAGTAGC (Trem2-rev), cctgctgaag-CAGGCTGGAGACGTGGAG (mKate2-fwd), and tttaaacgggccctcta-gacGGGCCCTCGAGTCACAGC (mKate2-rev). PCR products were purified by gel extraction before being cloned into a pcDNA3.1 vector.

### Transient transfection of HeLa cells
HeLa cells (CCL-2, ATCC) were grown in 6-well plates to subconfluency and transiently transfected with the TREM2-P2A-mKate2-KDEL construct in a pcDNA3.1 vector using X-tremeGENE DNA transfection reagent (Roche). 4 h after transfection media was exchanged. Cells were harvested at indicated time points and lysed in NP-40 STEN-lysis buffer (150 mM NaCl, 50 mM Tris–HCl pH 7.6, 2.5 mM ETDA, 1 % NP40) including protease inhibitor cocktail (Sigma, P8340). Lysates were centrifuged for 20 min at 17,000 g and 4 °C. The protein concentration of the soluble fraction was determined using the BCA protein assay (Interchim) and equal amounts of protein were separated by SDS-PAGE.

### SDS-PAGE and immunoblotting
Lysates were loaded onto precast 11-12% tris-glycine gels (5-7 μg and 15 μg per lane for HeLa and whole brain RIPA lysates, respectively) and separated by SDS-PAGE. In addition, we loaded SeeBlue™ Plus2 (Invitrogen) as molecular weight marker. Gels were run under reducing conditions for 90 min at 100 V. Subsequently, transfer onto PVDF membranes (Millipore, IPVH 00010) was carried out for 60 min at 400 mA. After blocking of membranes in either I block (ThermoFisher Scientific, /2015) or 5% milk (Roth, T145.2) in PBS for 60 min at RT primary antibodies (Supplementary Table 1) were added and membranes were incubated overnight at 4 °C. Membranes were washed three times for 10 min in TBS tween or TBS triton X-100. Next, HRP-conjugated secondary antibodies (Supplementary Table 1) were added, and membranes were incubated for 60 min at RT. After washing three times for 10 min membranes were incubated upside down in either Pierce™ ECL western blotting substrate (Thermo Scientific, 32106) or Pierce™ ECL plus western blotting substrate (Thermo Scientific, 32132 × 3). Development of membranes was finally conducted using x-ray films (Fuji Medical X-ray Films, 47410-19289) and an x-ray developer (Cawomat 2000 IR). For quantitative analysis, images were

taken by a Luminescent Image Analyzer (Fujifilm; ImageQuant; LAS-4000) and evaluated with the Multi Gauge software (Fujifilm; v3.0). TREM2 C-terminal fragments were separated using tris tricine gels (Invitrogen, Novex™, 10–20%), blotted onto nitrocellulose membrane (Amersham, 10600001) and boiled in PBS for 5 min before the blocking step.

## Genetic editing of mice

All animal experiments were approved by the Ethical Review Board of the Government of Upper Bavaria (animal licences ROB-55.2-2532.Vet_02-17-75, ROB-55.2-2532.Vet_02-22-125, ROB-55.2-2532.Vet_02-16-121, and ROB-55.2-2532.Vet_02-18-39). Mice were group housed with littermates on a normal 12-h light/dark cycle with *ad libitum* access to food and water. Both genders were used for all experiments. The mKate2 cassette was targeted to the area between exon 5 and the stop codon of the endogenous murine *Trem2* gene using a CRISPR/Cas9 strategy (Fig. 1E). For insertion of mKate2 into the *Trem2* locus, C57Bl/6 J mouse zygotes were injected with 40 ng/µl Cas9 protein (IDT), 17.5 ng/µl of annealed crRNA (5′-AGAAUUCUCU-CACGUACCUCGUUUUAGAGCUAUGCU-3′) and tracrRNA (IDT) and 10 ng/µl of donor vector containing the sequence encoding the self-cleaving peptide P2A, the cDNA of mKate2 and 5′ and 3′ homology regions (1000 bp each). Offspring was screened for the reporter cassette by PCR. Sequencing was performed on offspring with successful knock-in to ensure that there were no mutations in the insertion. Additionally, the most likely off target loci were sequenced before selecting the optimal animals to establish the colony. These off target loci were sequenced all of which were found not to be modified by the CRISPR procedure:

OT1: AGAACTTTCTCACGTACTTC CGG
OT2: AGATTTTTCTCAAGTACCTC AGG
OT3: AGATCTCACTCAGGTACCTC GGG
OT4: CGAAGTCCCTCACATACCTC CAG
OT5: ACAAGTCTCTCCTGTACCTC AGG
OT6: AAAAATCTCAAACGTACCTC CAG

Homozygous *Trem2-mKate* $^{KI/KI}$ mice were crossed with *APP/PS1 mice*[45] to generate transgenic *Trem2-mKate2*$^{KI/wt}$;*APP/PS1* (hereafter referred to as disease mice) as well as non-transgenic *Trem2-mKate2*$^{KI/wt}$ mice (hereafter referred to as healthy mice). For the experiments in which we dosed with the agonistic antibody, the reporter mice were crossed to *App*$^{SAA}$ knock-in[52] as well as *hTfR* knock-in (human transferrin receptor expressing) mice[55]. The experimental mice expressed the reporter heterozygous while *App*$^{SAA}$ and the *hTfR* knock-in were homozygous (*Trem2-mKate2*$^{KI/wt}$.*App*$^{SAA/SAA}$.*hTfR*$^{KI/KI}$). Dosing with either ATV:4D9[26] or the isotype control antibody (ATV:Iso) was started at 8 months. Animals received a total of four doses (1 mg/kg) i.p. where one dose was given each month (i.e., until an age of 11 months). One month after the final dose mice were sacrificed (12 months of age). Two cohorts of mice were treated to enable sorting of the subpopulations by FACS as well as collection of CSF and brain tissue for biochemistry and histology. All mice were on C57BL/6 J background.

## Traumatic brain injury

Controlled cortical impact (CCI), a model of brain injury was performed as described previously[42,86]. Briefly, 8-week-old male *Trem2-mKate2* reporter mice were injected with buprenorphine (0.1 mg/kg) before being anesthetized with isoflurane (4%, 3 seconds). The CCI was achieved by using a pressure-controlled impactor (L. Kopacz, University of Mainz) with the following settings: impact velocity 6.5 m/s, 0.5 mm penetration depth and 150 ms contact time. The injury was delivered directly to the dura after a right parietal craniotomy had been performed. The exposed dura was sealed using tissue glue (VetBond™, 3 M Animal Care Products). During the whole procedure, anaesthesia was maintained with 1.5-2.5% isoflurane in oxygen/air and the animals heart rate and body temperature was monitored

continuously. After the surgery, animals were allowed to recover in a heating chamber at 34 °C with 60% humidity, to avoid hypothermia. Mice were monitored, scored and injected with carprofen (5 mg/kg) for 3 days following the surgery. 72 h post impact animals were sacrificed and perfused, alongside naïve animals which had not undergone the procedure.

## CSF collection

CSF collection was performed as described elsewhere[87]. Briefly, mice were deeply anesthetized using a mixture of medetomidine [0.5 mg/kg], midazolam [5 mg/kg], and fentanyl [0.05 mg/kg] (MMF). Once toe pitch reflex was absent, mice were fixed into a stereotactic frame. The cisterna magna was exposed and the dura punctured with a borosilicate glass capillary (Sutter). A medical grade tubing was connected to the glass capillary. Utilizing a syringe at the end of the tubing to generate backpressure, the CSF was slowly collected. CSF was ejected into protein LoBind tubes (Eppendorf, 0030108094) on ice. Samples were centrifuged at 2000 g for 10 min at 4 °C to check for contamination by red blood cells. Upon visual inspection, 5 µl CSF was aliquoted in protein LoBind tubes and snap frozen in liquid nitrogen.

## Perfusion

At indicated ages mice were anesthetized by injection of Ketamine/Xylazine mix or $CO_2$ inhalation before being transcardially perfused with ice-cold PBS. The brain was removed and separated into hemispheres. One half was snap frozen for biochemical analysis and the other half was post fixed by immersion in cold 4% paraformaldehyde (PFA) for histological assessment. For FACS analysis, mice were perfused with PBS, olfactory bulb and cerebellum were removed and the remaining brain was used to isolate microglia. At indicated times, mice received a single dose of 30 mg/kg FDG (Sigma) 1 h prior take down (see Fig. 7).

## Immunofluorescence

Mouse tissue was changed from 4% PFA to PBS the day after perfusion. 50 µm thick sagittal brain sections were cut on a vibratome, collected in PBS and stored at 4 °C until needed. Sections were transferred to 24-well plates and incubated with a blocking buffer (BB) containing 0.5% BSA and 0.5% TritonX-100 in 1x PBS, for 1 h at room temperature (RT) with slow agitation. Primary antibodies (Supplementary Table 1) were diluted in BB and incubated overnight at 4 °C again with slow agitation. The next day sections were washed 3 × 10 min in 1x PBS before Alexa conjugated secondary antibodies (1:500) and DAPI (1:1000) (Supplementary Table 1), diluted in BB were added. Sections were incubating in the dark for 1 h at RT while shaking. Sections were washed again 3 × 10 min before being mounted on glass slides and cover slipped using Fluoromount-G (Invitrogen). Slides were sealed with clear nail polish and allowed to dry for at least 24 h in the dark before being imaged.

HeLa cells (CCL-2, ATCC) were grown on cover slips and transiently transfected with lipofectamine 2000 (Invitrogen). After 48 h, cells were washed 3x with PBS before being fixed with 4% PFA for 20 min at RT. Cells were permeabilized for 10 min at RT with PBS containing 0.2% Triton-X. Cells were washed in PBS 3 × 10 min and then blocked in 5% serum for 1 h at 37 °C. Primary antibody (Supplementary Table 1) was diluted in BB and incubated at RT for 2 h. After washing, cells were incubated with secondary antibodies and DAPI for 1 h at RT. Cells were washed once more before being mounted on glass slides. Images were taken with a LSM800 (Zeiss) Confocal microscope. Laser power and gain were kept consistent across imaging sessions. The Zen blue software (Zeiss) or ImageJ were used for quantification.

## RT-qPCR

Total RNA isolation was performed using the RNeasy Mini kit (Qiagen), following the manufacturer's instructions. Briefly, either harvested

cells or mouse brain powder was resuspended in RLT buffer (Qiagen) for cell lysis and transferred onto QIAshredder columns (Qiagen). Optimal DNase I treatment was performed for 15 minutes at RT. RNase-free water was used to elute the RNA and concentrations measured using a Nanodrop. RNA was kept at -80 °C or directly used for cDNA generation using 2 µg of RNA following the Superscript IV reverse transcriptase protocol (Invitrogen) with random primers (Promega). TaqMan technology with the following primer sets from Integrated DNA Technologies was used: Trem2 Mm.PT.58.46092560, Clec7a Mm.PT.58.42049707, Cd68 Mm.PT.58.32698807, Grn Mm.PT.58.31751824. For mKate2, 5′-CAACTTCAGATCCTCACTGGAC-3′ and 5′-TTAATCAGCTCGCTCACCATAG-3′ were used as forward and reverse primers, respectively. Likewise, 5′-/56-FAM/AACGGGAAG/ZEN/CGGAGCTACTAACTT/3IABkFQ/-3′ was used as a probe. For the Trem2 WT allele, 5′-CTGGACTGTGGCCAAGATG-3′ and 5′-GCTGGACT-TAAGCTGTAGTTCTC-3′ were used as forward and reverse primers, respectively. Likewise, 5′-/56-FAM/TGGGCACCA/ZEN/ACTTCA-GATCCTCAC/3IABkFQ/-3′ was used as a probe. RNA expression was measured in triplicates using TaqMan assays on a 7500 Fast Real-Time PCR System (Applied Biosystems) normalized to Hprt Mm.PT.39a.22214828. Relative transcription levels of the respective sequences were analyzed using the comparative delta Ct method (7500 Software V2.0.5, Applied Biosystems).

## Microglia isolation for FACS

Microglia isolation was performed using the MACS system (Miltenyi Biotec) as described previously[88]. Briefly, mice were perfused with ice-cold 1x PBS as mentioned above. The brain was carefully removed and placed in cold DPBS. The cerebellum was discarded and the rest of the brain cut into small pieces before being transferred to gentleMACS C tubes (Miltenyi Biotec, 130-093-237) for enzymatic digestion using the gentleMACS Octo Dissociator (Miltenyi Biotec, 130-134-029). The Adult Brain Dissociation Kit (Miltenyi Biotec, 130-107-677) was used following manufacturer instructions. Following dissociation, the samples were strained, washed and centrifuged to generate a single cell suspension. Optional debris removal was performed following the manufacturer's instructions. No CD11b enrichment step was performed as the intrinsic mKate2 reporter is only present in TREM2 expressing (microglia) cells. Hence, FACS was performed based on the mKate2 fluorescence signal and sorted into low, mid and high expressing subpopulations, as a proxy for the TREM2 expression of individual cells. The sorted fractions were spun down and the pellets stored at -80 °C before being processed for further analysis, as described below.

## Transcriptomic analysis – *APP/PS1* mice

For a subpopulation of microglia RNA sequencing was performed. To this purpose, cells were sorted as described above and immediately stored in Trizol. RNA extraction was performed using the RNeasy Micro kit (Qiagen) following manufacturer's instructions. For RNA sequencing, both the RNA quantity and integrity were assessed using the HS RNA assay on a Tapestation 4200 system (Agilent). The Smart-seq2 protocol, as described by Picelli et al.[89], was used for the generation of non-strand-specific, full transcript sequencing libraries. Briefly, 1 ng of total RNA was transferred to buffer containing 0.2% TritonX-100, protein-based RNase inhibitor, dNTPs, and oligo-dT oligonucleotides to prime the subsequent RT reaction on polyadenylated mRNA sequences. The SMART RT reaction was conducted at 42 °C for 90 min using commercial SuperScript II (Invitrogen) and a Template-Switching Oligo. A pre-amplification PCR of 16 cycles was carried out to generate double-stranded DNA from the cDNA template. Subsequently, 100 pg of amplified cDNA were used for tagmentation and enrichment using the Nextera XT kit (Illumina) to construct the final sequencing libraries. Libraries were quantified using the Qubit HS dsDNA assay, and library fragment size distribution was determined

using the D1000 assay on a Tapestation 4200 system (Agilent). The sequencing was performed in single-end mode (75 cycles) on a Next-Seq 500 System (Illumina) with NextSeq 500/550 High Output Kit v2.5 (150 Cycles) chemistry. Raw sequencing data were demultiplexed using bcl2fastq2 (v2.20). The sequenced reads were aligned against the Gencode mouse reference genome vM16 using kallisto (0.44.2)[90] and samples with more than 50,000,000 reads were downsampled to 50 % of the original number of reads. The following analysis steps were performed in R (v4.1.0)[91] and R Studio (v1.4.1717)[92]. The count matrix was imported into the analysis pipeline using R/tximport (v1.20.0)[93]. Genes with less counts than number of samples were excluded from the analysis resulting in 26,492 genes kept in the analysis. Normalization of the count matrix was computed with R/DESeq2 (v1.32.0)[94] and a variance stabilizing transformation applied using the DESeq2 rlog function at default settings. Differential expression analysis based on the DESeq2 package was performed adjusting p-values according to independent hypothesis weighting from the IHW package (v1.20.0)[95] and applying empirical Bayes shrinkage estimators from the R/apeglm package (v1.14.0)[96]. DEGs were defined based on a fold change threshold > 2 and a p-value threshold of < 0.05. Cell type abundances of the variance stabilized data were determined by CIBERSORTx (https://cibersortx.stanford.edu/)[97] with default parameters using two microglia single-cell RNA-seq data sets (GSE98969[8], GSE165306[48]) as reference. Signature enrichment was calculated on the variance-stabilized data per sample using R/GSVA (v1.40.1)[98] with default settings and a Wilcoxon rank sum test with subsequent Benjamini-Hochberg adjustment was applied with R/rstatix (v0.7.0, https://CRAN.R-project.org/package=rstatix) to test for significance. Permutation tests were computed by drawing 500 random, unique signatures of the same size as the signature being assessed. Gene set variation analysis (GSVA) enrichment scores were calculated for each gene signature and an ANOVA followed by Benjamini-Hochberg adjustment was applied over the 5 microglia subpopulations. The likelihood of the enrichment result of the tested signature was computed by dividing the number of random gene signatures with lower adjusted p-values by the number of permutations. To assess the signature enrichment on a population level, expression level statistics from a non-parametric kernel estimation of the cumulative density function of each variance-stabilized gene expression profile per sample were calculated[98] prior to computing the mean for the 5 different microglia subpopulations. A gene set enrichment analysis (GSEA) was performed with the transformed data as the input using R/fgsea (v1.18.0)[99].

The gene co-expression network analysis was performed using R/hCoCena (v1.0.0)[100,101] on the normalized count matrix. Gene pairs with a Pearson's correlation coefficients lower than 0.879 were excluded resulting in a network with 4259 genes, 207,764 edges and an $R^2$-value of 0.748. The Leiden clustering algorithm was used to identify 6 gene modules with a minimum size of 50 genes containing a total of 4,002 genes. Gene Ontology (GO)[102,103] and the Molecular Signature Database (MSigDB)[104,105] Hallmark gene set enrichment was performed on the module genes making use of the functions included in the hCoCena workflow with default parameters. The bulk RNA-seq data of the *APP/PS1* mice is available under the following GEO accession number: GSE271074

## Transcriptomic analysis – *App^SAA* mice

For one cohort of microglia treated with either ATV:4D9 or ATV:Iso, RNA sequencing was performed. RNA isolation was performed using the RNeasy Micro kit (Qiagen), following manufacturer's instruction. For the RNA sequencing, both the quantity and integrity of RNA were assessed using the HS RNA assay on a Tapestation 4200 system from Agilent. The Smart-seq2 protocol, as detailed by Picelli et al.[89], was employed to create non-strand-specific, full transcript sequencing libraries. In summary, 2 µL of total RNA were transferred to a buffer containing 0.2% TritonX-100, protein-based RNase inhibitor, dNTPs,

and oligo-dT oligonucleotides for priming the subsequent RT reaction on polyadenylated mRNA sequences. The SMART RT reaction was conducted at 42 °C for 90 min using commercial SuperScript II (Invitrogen) and a Template-Switching Oligo. The pre-amplification PCR with 17 cycles was carried out to produce double-stranded DNA from the cDNA template. After normalization, 100-300 pg of amplified cDNA underwent tagmentation and enrichment using self-loaded Tn5 transposase (Diagenode) to construct the final sequencing libraries. Quantification of libraries was performed using the Qubit HS dsDNA assay, and the distribution of library fragment sizes was determined using the D1000 assay on a Tapestation 4200 system (Agilent). The sequencing was executed in single-end mode (75 cycles) on a NovaSeq 6000 System (Illumina) with a NovaSeq 6000 S2 Reagent Kit v1.5 (300 cycles) chemistry. Raw sequencing data were demultiplexed using bcl2fastq2 (v2.20) and aligned against the Gencode mouse reference genome vM16 using kallisto (v0.48.0)[90]. The following analysis steps were performed in R (v4.3.0)[106] and R Studio (v1.4.1717)[92]. The count matrix was imported into the analysis pipeline using R/tximport (v1.28.0)[93]. Genes with less than 10 counts in at least the minimal number of samples in the smallest condition, 4 samples, were excluded from the analysis resulting in 27,016 genes kept in the analysis. Normalization of the count matrix was computed with R/DESeq2 (v1.40.2)[94] and a variance stabilizing transformation applied using the DESeq2 vst function at default settings. Statistical analysis of the gene expression differences was done with the Wilcoxon test followed by a Benjamini-Hochberg correction for multiple testing from the package R/ rstatix (v0.7.2, https://CRAN.R-project.org/package=rstatix). The bulkRNAseq data of the $App^{SAA}$ mice is available under the following GEO accession number: GSE295016

## Analysis of human post-mortem brain microglia single-nucleus transcriptome data

To verify our findings linking plaque proximity to increased Trem2 expression levels in bulk RNA-seq data of the *APP/PS1* mice, we inspected human post-mortem brain single-nucleus transcriptome data by Gabitto et al.[47] The microglia/PVM subset from the dorsolateral prefrontal cortex and middle temporal gyrus were merged resulting in a total of 82,486 cells and subsequently analyzed using Seurat (v4.0.4)[107]. First, the gene expression values were normalized by multiplying the total UMI counts per cell by 10,000 (TP10K) and applying a natural log transformation by log(TP10K + 1). Subsequently, the normalized data was scaled and centred, and the 2000 most variable genes were identified following a variance-stabilization using the 'vst' method. Based on these features, a PCA was performed of which the first 12 PCs were utilized to calculate a shared nearest neighbour (SNN) graph. Cells were clustered based on the SNN graph using the Louvain algorithm at a resolution of 0.1. Cluster identities were determined using the Seurat FindAllMarkers function using the Wilcoxon rank sum test defining cluster marker genes as genes expressed in at least 20 % of the cells of either of the two compared populations, with a difference of at least 10 % between the populations, and a log fold change > 0.5. Subsequently, cells were annotated in a two-step process by overlapping cluster marker genes and protein marker expression with literature-known cell-type marker genes. Next, all microglia from donors with no or high Alzheimer's disease neuropathological change (ADNC) were selected, resulting in a total of 46,345 microglia, and genes expressed in less than 5 cells were excluded prior to re-normalization and scaling as described above. Following a variance-stabilizing transformation, a PCA of all microglia was performed using the 2000 most variable genes applying the 'vst' method. To create a two-dimensional representation of the data, a SNN graph of the first 8 PCs was computed, and the resulting nearest neighbours were then used to calculate a uniform manifold approximation and projection (UMAP). To infer the plaque proximity, the PIG signature by Chen et al.[46] was enriched in microglia using the AddModuleScore function

and the inferred plaque proximity was then modelled against the TREM2 level in TREM2-expressing cells. To test for differences in the models of the ADNC groups, a likelihood ratio test (LRT) was performed using the lme4[108] (v1.1-27.1) and lmtest[109] package (v0.9-38) comparing a linear mixed effect model predicting the plaque proximity using the TREM2 expression level per ADNC group to a null model setting the donor identity as a random variable.

## Single cell radiotracing

Animals (14-month-old) were food deprived for 2–3 h before the experiment started. Mice received a single tail vein injection of $45 \pm 3$ MBq [$^{18}$F]fluorodeoxyglucose (FDG)[25,53]. 30 min post injection animals were sacrificed by cervical dislocation and their brains removed. A single cell suspension was generated using the MACS system as described above ("Microglia isolation for FACS") and previously[53]. Microglia were incubated with 10 μl of CD11b MicroBeads (Miltenyi Biotec) per $10^7$ total cells for 15 min at 4 °C. Cells were washed, centrifuged at 300 g for 10 min, the pellet resuspended in 500 μl of D-PBS−0.5% BSA and then added to pre-wetted LS columns (Miltenyi Biotec). The columns were washed with $3 \times 3$ ml of D-PBS−0.5% BSA buffer after which the cells were flushed out using 5 ml of D-PBS−0.5% BSA buffer. The suspension was centrifuged, and the remaining cell pellet resuspended in 1 ml of cold D-PBS and 10 μl DAPI staining solution. Using a MoFlo Astrios EQ cell sorter (B25982, Beckman Coulter), microglia were sorted based on their mKate2 expression into the subpopulations as described in the previous section. Sorted cell pellets were analysed in a gamma counter (Hidex AMG Automatic Gamma Counter) with decay correction to time of tracer injection for final activity calculations. The measured radioactivity (Bq) of the cell pellet was divided by the specific cell number in the pellet, to calculate radioactivity per single cell. Radioactivity per cell was then normalized by injected radioactivity and body weight[110].

## Metabolomic and Lipidomic analyses

Metabolomic and lipidomic analyses were performed as described previously[52], with some modifications. Sample extraction: Pellets from the sorted subpopulations were reconstituted on ice in 9:1 MeOH:water including internal standards, vortexed for 1 min, and spun down for 5 min at 10,000 g. Supernatant was transferred to glass vial for analysis by LC-MS. Polar metabolites in electrospray positive mode: Targeted analysis of 275 polar metabolites was performed on an Agilent 1290 UPLC system (Agilent) coupled to Sciex triple quadrupole mass spectrometer (QTRAP 6500 + , Sciex). All other settings were as described in Xia et al.[52] Metabolites analyses were performed liquid chromatography (Agilent 1290 UPLC system) coupled to electrospray mass spectrometry (QTRAP 6500 + , Sciex). For each analysis, 5 μL of sample was injected on a BEH amide 1.7 μm, $2.1 \times 150$ mm column (Waters Corporation, Milford, Massachusetts, USA) using a flow rate of 0.40 mL/min at 40 °C. Mobile phase A consisted of water with 10 mM ammonium formate + 0.1% formic acid. Mobile phase B consisted of acetonitrile with 0.1% formic acid. The gradient was programmed as follows: 0.0–1.0 min at 95% B; 1.0–7.0 min to 50% B; 7.0–7.1 min to 95% B; and 7.1–10.0 min at 95% B. The following source settings were applied: curtain gas at 30 psi; collision gas was set at 8 psi; ion spray voltage at 5500 V; temperature at 600 °C; ion source Gas 1 at 50 psi; ion source Gas 2 at 60 psi; entrance potential at 10 V; and collision cell exit potential at 12.5 V. Polar metabolites in electrospray negative mode: Samples were dried and reconstituted in same volume of 9:1 MeOH:water fortified with 0.1% formic acid. Acidified samples were chromatographically resolved on an Imtact Intrada Organic Acid column (3 μm, $2 \times 150$ mm, Imtakt) using a flow rate of 0.2 ml/min at 60 °C. Mobile phase A consisted of acetonitrile/water/formic acid = 10/90/0.1%. Mobile phase B consisted of acetonitrile/ 100 mM ammonium formate= 10/90%. The gradient was programmed as follows: 0.0–1.0 min at 0% B; 1.0–7.0 min to 100% B; 7.1 at 0% B; and 7.1–10 min

at 0% B. Source settings used were as follows: curtain gas at 40 V; collision gas was set at medium [collision gas for TQ 6500+ settings was set at 8 psi]; ion spray voltage at -4500 V; temperature at 600 °C; ion source Gas 1 at 50 psi; ion source Gas 2 at 60 psi; entrance potential at -10 V; and collision cell exit potential at -15.0 V. Lipidomics in positive and negative ionization modes: Targeted analysis of 153 lipids were performed as described in Xia et al.[52] Lipid analyses were performed by liquid chromatography (Agilent 1290 UPLC system), coupled to electrospray mass spectrometry (QTRAP 6500 + , Sciex). For each analysis, 5 μL of sample was injected on a BEH C18 1.7 μm, 2.1 × 100 mm column (Waters) using a flow rate of 0.25 mL/min at 55 °C. Electrospray ionization was performed in positive and negative ion modes. For positive ionization mode, mobile phase A consisted of 60:40 acetonitrile/water (v/v) with 10 mM ammonium formate + 0.1% formic acid; mobile phase B consisted of 90:10 isopropyl alcohol/acetonitrile (v/v) with 10 mM ammonium formate + 0.1% formic acid. For negative ionization mode, mobile phase A consisted of 60:40 acetonitrile/water (v/v) with 10 mM ammonium acetate + 0.1% acetic acid; mobile phase B consisted of 90:10 isopropyl alcohol/acetonitrile (v/v) with 10 mM ammonium acetate + 0.1% acetic acid. The gradient was programmed as follows: 0.0-8.0 min from 45% B to 99% B, 8.0-9.0 min at 99% B, 9.0-9.1 min to 45% B, and 9.1-10.0 min at 45% B. Electrospray ionization was performed using the following settings: curtain gas at 30 psi; collision gas at 8 psi; ion spray voltage at 5500 V (positive mode) or -4500 V (negative mode); temperature at 250 °C (positive mode) or 600 °C (negative mode); ion source Gas 1 at 55 psi; ion source Gas 2 at 60 psi; entrance potential at 10 V (positive mode) or -10 V (negative mode); and collision cell exit potential at 12.5 V (positive mode) or -15.0 V (negative mode). Data acquisition and analysis: Data acquisition was performed using Analyst 1.6.3 (Sciex) in multiple reaction monitoring mode (MRM) with optimized collision settings. Peak integration and quantification were performed using MultiQuant 3.02 (Sciex) software. Endogenous metabolites/lipids were expressed as area ratios to specific spiked in stable isotope internal standards. Peak annotation and integration were based on defined retention times and MRM properties of commercial reference standards. Of the 428 targeted analytes, 141 analytes passed QC, missing values, and outlier cheques. Based on replicate analysis of pooled QC samples, mean coefficient of variance for across all analytes was 16.4%. For all analysis, raw area ratio data was log2 transformed and filtered to retain variables that are present in at least 70% of samples. Missing values were imputed using the kNN method using the 5 nearest donor variables. Unwanted variations in the dataset were adjusted for using the RUV4 method. Assessment of genotype (healthy vs disease) and TREM2 status (low, mid, high), a group factor was created by combining genotype and TREM2 expression level status. A linear mixed model was conducted with individual mice as a random effect, group status as independent factor, and RUV4 regression factors as covariates. Benjamini-Hochberg adjustment was used to adjust for multiple comparisons. Top contrast results were visualized using heatmaps, boxplots or volcano plots. All analyses were performed using R statistical software (version 4.0.2; R Core Team 2020).

## Myelin isolation and purification

Myelin was isolated from 12-week-old mouse brains (C57/BL6). Brains were homogenized in 0.32 M saccharose solution, added gently on top of 0.85 M saccharose solution in an ultracentrifugation bucket and centrifuged for 30 min at 75,000 g. The interphase, with roughly purified myelin, was removed, washed with water and centrifuged for 15 min at 75,000 g. Osmotic shock was performed by water incubation for 15 min followed by centrifugation for 15 min at 12,000 g. The pellet was resuspended in 0.35 M saccharose solution, added on top of 0.85 M saccharose solution and centrifuged for 30 min at 75,000 g. Purified myelin was washed with water and centrifuged for 15 min at 75,000 g. The pellet was resuspended in PBS, homogenized and stored

at −80 °C. Protein concentration of myelin was measured by Bradford assay.

## Myelin labelling

Myelin was labelled with pHrodo-green according to the manufacturer's specifications (ThermoFisher Scientific, P36015) with the following modifications. 100 μl myelin at 1 mg/ml was mixed with 10 μl 1 M sodium bicarbonate in protein LoBind tubes (Eppendorf, 0030108116). Labelling of myelin was carried out for 45 min at RT in the dark. After addition of 100 μl PBS labelled myelin was pelleted (12,000 g; 10 min, 4 °C). Myelin was washed with PBS two more times to remove all excess pHrodo. Finally, myelin was resuspended in 100 μl PBS and frozen at -80 °C until further use.

## Ex vivo phagocytosis assay

The ex vivo phagocytosis assay to quantify the percentage of pHrodo-positive microglia was conducted as described before[26]. In brief, brains from *Trem2-mKate2^{KI/wt};APP/PS1* and *Trem2-mKate2^{KI/wt}.App^{SAA/SAA}.hTfR^{KI/KI}* mice were collected as described above. Likewise, single cell suspensions were generated using the Miltenyi system (103-107-677) as mentioned previously. In the following step, 50 μl cell suspension was incubated for 20 min with CD11b-BV421, CD45-APC and mouse Fc blocker (Supplementary Table 1) at 4 °C. Cells were washed once in FACS buffer containing 1% BSA (Miltenyi, 130-091-376) and 1 mM EDTA in PBS. Microglia concentration in each sample was determined with a Cytek Northern Lights (Cytek Biosciences). Brain cell suspensions were diluted to a concentration of 500 microglia per microliter. 100 μl brain cell suspension was mixed with 100 μl pHrodo-labelled myelin at 80–100 μg/ml on ice in 0.5% BSA (Miltenyi, 103-091-376) in DPBS. Phagocytosis of pHrodo-labelled myelin was carried out at 37 °C for 45–60 min (one sample was kept at 4 °C as a negative control). The samples were gently flicked every 15 min. Following the incubation, 1 ml 0.5% BSA in DPBS was added to each sample, cells spun down (500 g, 5 min, 4 °C) and stained in 200 μl FACS buffer (CD11b-BV421, CD45-APC, mouse Fc blocker) for 20 min at 4 °C. After the addition of 1 ml FACS buffer cells were spun down again (500 g, 5 min, 4 °C), resuspended in FACS buffer and strained (Corning, 352235). FACS analyses were carried out using a BD FACSAria Fusion (BD Biosciences). 50,000 cells and 200,000 cells per sample were analyzed for *Trem2-mKate2^{KI/wt};APP/PS1* and *Trem2-mKate2^{KI/wt}.App^{SAA/SAA}.hTfR^{KI/KI}* mice, respectively. Quantification of percentages of pHrodo-positive cells and median fluorescence intensity in the low, mid and high mKate2 subpopulations was performed using FlowJo software.

## Generation of RIPA extracts from mouse brain

Frozen brain hemispheres were pulverized using a liquid nitrogen cooled mini mortar (Sigma, Z756377). 10-20 mg of brain powder was transferred to soft tissue homogenizing Precellys tubes (Bertin Technologies, P000933-LYSK0-A.0) on dry ice. 250 μL freshly prepared RIPA buffer (Pierce, 89901) supplemented with 1 mM EDTA and protease inhibitor cocktail (Sigma, P8340) was added to each sample on ice. Samples were subsequently homogenized at 6500 rpm for 30 sec at 4 °C using a sample homogenizer (Bertin Technologies, Precellys Evolution). Next, tubes were shaken at 400 rpm for 10 min at 4 °C to continue with homogenization followed by centrifugation (5000 g for 10 min at 4 °C). The supernatant was then subjected to ultracentrifugation (186,000 g for 60 min at 4 °C) to generate RIPA soluble extracts.

## Proteomics

CSF samples were processed as previously described[111]. Briefly, ammonium bicarbonate (50 mM) and sodium deoxycholate (0.1%) were added. Disulphide bonds were reduced by addition of 2 μl 10 mM dithiothreitol (DTT, Biomol) and incubation for 30 min at 37 °C, followed by alkylation by addition of 2 μl 55 mM iodoacetamide (IAA,

Sigma Aldrich) and incubation for 30 min at RT in the dark. Proteins were digested sequentially with LysC (0.1 μg, 4 h; Promega) and trypsin (0.1 μg, 16 h; Promega) at RT. Samples were acidified with 4 μl of 8% formic acid (Sigma Aldrich) and 150 μl of 0.1% formic acid (Sigma Aldrich). Precipitated deoxycholate was removed by centrifugation at 16,000 g for 10 min at 4 °C. Peptides were desalted by stop and go extraction (STAGE) with C18 tips. The samples were then vacuum-dried and reconstituted in 20 μL of 0.1% formic acid using a sonication bath before mass spectrometric analysis.

Brain samples were digested using the single-pot, solid-phase-enhanced sample preparation (SP3)[112] with slight modifications. Briefly, 20 μg of brain lysates were diluted 1:2 with $H_2O$, $MgCl_2$ (10 mM) and 25 units benzonase (Sigma-Aldrich) were added. DNA digestion was performed for 30 min at 37 °C. Proteins were reduced by using 15 mM DTT for 30 min at 37 °C, followed by cysteine alkylation with 60 mM IAA for 30 min at 20 °C. Excess IAA was removed by adding DTT. After binding of proteins to 40 μg of a 1:1 mixture of hydrophilic and hydrophobic magnetic Sera-Mag SpeedBeads (GE Healthcare) with a final concentration of 70% acetonitrile for 30 min at RT, beads were washed four times with 200 μL 80% ethanol. Digestion was performed for 16 h with 0.25 μg LysC and 0.25 μg trypsin (Promega). Supernatants were filtered wit 0.22 μm spin-filters and dried by vacuum centrifugation. Peptide concentrations were estimated using the Qubit protein assay (Thermo Fisher).

The LC-MS/MS proteomic analyses of CSF samples was performed on a nanoElute system (Bruker Daltonics) which was online coupled with a timsTOF pro mass spectrometer (Bruker Daltonics). A volume of 4 out of 20 μL for CSF samples were separated on a self-packed 15 cm C18 column (75 μm ID) packed with ReproSil-Pur 120 C18-AQ resin (1.9 μm, Dr. Maisch GmbH) using a binary gradient of water and acetonitrile (ACN) supplemented with 0.1% formic acid with a flow rate of 300 nL/minute (0 min, 2% B; 2 min, 5% B; 62 min, 24% B; 72 min, 35% B; 75 min, 60% B) and a column temperature of 50 °C. The nanoHPLC was online coupled to a TimsTOF pro mass spectrometer (Bruker) with a CaptiveSpray ion source (Bruker). A Data Independent Acquisition Parallel Accumulation–Serial Fragmentation (diaPASEF) method was used for spectrum acquisition. Ion accumulation and separation using Trapped Ion Mobility Spectrometry (TIMS) was set to a ramp time of 100 ms. One scan cycle included one TIMS full MS scan with 26 windows with a width of 27 m/z covering a m/z range of 350-1001 m/z. Two windows were recorded per PASEF scan. This resulted in a cycle time of 1.4 seconds.

Brain samples were analyzed on a Vanquish Neo coupled online via a Sonation column oven to an Orbitrap Exploris 480 mass spectrometer (Thermo Fisher Scientific). A peptide amount of 400 ng was separated with a two-column setup using an Acclaim Pepmap trapping column (5 μm, 0.3 × 5 mm) with a Pepsep separation column (15 cm × 75 μm × 1.9 μm, Bruker). Peptides were separated using a binary gradient of water and 80% acetonitrile (ACN) supplemented with 0.1% formic acid with a flow rate of 300 nL/minute (0-min, 6% B; 2 min, 6% B; 112 min, 31% B; 120 min, 50% B; 126.5 min, 95% B) and a column temperature of 50 °C.

A Data Independent Acquisition method was used. Peptide spectra were acquired at a resolution of 120,000 using an automatic gain control (AGC) of 300% followed by peptide fragmentation with 38 variable windows overlapping 1 m/z covering a m/z range of 350–1001 (m/z window widths: 34, 32, 25, 22, 20, 19, 19, 18, 17, 17, 17, 16, 16, 15, 15, 14, 14, 14, 14, 14, 15, 15, 16, 16, 17, 17, 17, 18, 18, 18, 19, 19, 22, 22, 24, 25, 32, 35). Peptide fragmentation spectra were acquired with an AGC of 3000%, normalized collision energy of 27.5% ± 10%, and automatic maximum injection time.

The software DIA-NN version 2.02 was used to analyze the data[113]. The raw data was searched against an in silico generated spectral library using a one protein per gene database from *Mus musculus* (UniProt, 21765 entries, download: 2025-02-04) and a database with

potential contaminants (233 entries) as well as the mKate2 and TREM2 sequences. Trypsin was defined as protease and two missed cleavages were allowed. Oxidation of methionines and acetylation of protein N-termini were defined as variable modifications, whereas carbamidomethylation of cysteines was defined as fixed modification. Variable modifications were restricted to two per peptide. The precursor and fragment ion m/z ranges were limited from 350 to 1001. Precursor charge states were set to 2-4 charges for the CSF data and 1–4 for the brain data. An FDR threshold of 1% was applied for peptide and protein identifications. The mass accuracy for MS1 and MS2 were set to 15 ppm. Ion mobility windows were automatically adjusted by the software. The match between runs and RT-dependent cross-run normalization options were enabled.

The software Perseus version 1.6.2.3[114] was used for further data analysis. The contaminants were removed. Protein label-free quantification (LFQ) intensities were log2 transformed. Changes in protein abundances between the different groups were calculated and a two-sided Student's T-test was applied comparing the log2 transformed LFQ intensities between the groups. At least three valid values per group were required for statistical testing. To account for multiple hypotheses, a permutation-based FDR correction[115] was applied.

## Statistics

Statistical analysis was carried out using GraphPad Prism software. Student's t-test (unpaired, two-tailed) was used to compare two groups, while one-way ANOVA (with Tukey's post-hoc analysis) was used to compare multiple groups. A value of $p < 0.05$ was considered statistically significant. In all Figures, N numbers correspond to biological replicates except for Fig. 1C,D where data from technical replicates are displayed. Statistical analyses of the Omics data were performed as described in the respective methods sections.

## Reporting summary

Further information on research design is available in the Nature Portfolio Reporting Summary linked to this article.

## Data availability

All data analysed and interpreted in this study are included in this published article or its supplementary files. The RNA-seq data generated in this study have been deposited under GEO accession codes GSE271074 (*APP/PS1* mice) (https://www.ncbi.nlm.nih.gov/geo/query/acc.cgi?acc=GSE271074) and GSE295016 (*App^SAA* mice) (https://www.ncbi.nlm.nih.gov/geo/query/acc.cgi?acc=GSE295016). The proteomic data generated in this study have been deposited to the ProteomeXchange Consortium via the PRIDE partner repository under accession code PXD065438[116]. The LC/MS data generated in this study have been deposited in the Metabolomics Workbench database under accession codes ST003501 (*APP/PS1* mice) and ST004230 (*App^SAA* mice) [Project https://doi.org/10.21228/M83F9Q]. The sc/snRNA-seq data used in this study are available under GEO accession codes GSE98969 and GSE165306 and at CELLxGENE (https://cellxgene.cziscience.com/collections/1ca90a2d-2943-483d-b678-b809bf464c30). Source data are provided with this paper.

## Code availability

All original code to reproduce key steps of the RNA-seq analysis has been deposited at GitLab under (https://gitlab.dzne.de/ag-ulas/Trem2-reporter-mouse-model) and Zenodo under https://doi.org/10.5281/zenodo.17793705.

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

## Acknowledgements

This work was supported by the Koselleck Project HA1737/16-1 (to CH) of the Deutsche Forschungsgemeinschaft (DFG, German Research Foundation) and the Excellence Strategy within the framework of the Munich Cluster for Systems Neurology (EXC 2145 SyNergy – ID 390857198). ARe was supported by a Ph.D. stipend from the Hans and Ilse Breuer Foundation. IK was supported by DFG grant 457586042. SFL was supported by the Alzheimer Forschung Initiative e.V. (#24013 R) with kind support of the Stiftung Alzheimer Initiative (SAI) and by the Centres of Excellence in Neurodegeneration for grant CoEN6005. The authors thank the animal facility staff and management of the Centre for Stroke and Dementia Research for excellent animal care. We thank Jamal Alkabsh for technical assistance with mass spectrometry data generation. We thank Pardis Khosravani as well as the Core Facility Flow Cytometry at the Biomedical Centre, Ludwig-Maximilians Universität München and the iFlow Core Facility of the University Hospital LMU Munich (INST 409/225-1 FUGG) for technical assistance with cell sorting. We thank Dr. Michael Willem for critical reading of the manuscript. Some figure schematics were created using BioRender and were used with permission.

## Author contributions

C.H. designed the study and applied for funding. A.F.F., K.D., K.S., A.Re., B.W., M.Br., K.M.M., A.C., J.L.S. and C.H. designed experiments. A.F.F., K.D., K.S., A.Re., J.H.S., Bv.L., B.W., L.M.B., K.W., L.W., T.U., E.D., P.G., S.P., I.K., A.Ri., B.T., T.H., S.A.M., B.B., C.K., K.B., B.N., L.S., N.J., S.A.B., S.S.D. performed experiments. T.U., W.W., N.P., M.Br., M.Be., W.W., M.S., G.D.P., J.W.L., K.M.M., A.L., A.C., J.L.S. and C.H. supervised the study. A.F.F., K.D., K.S., A.Re., J.H.S., Bv.L., L.M.B., L.W., P.G., B.T., T.H., A.Ri., M.Br., G.D.P., J.W.L., K.M.M., A.L., A.C., S.A.M., S.S.D., S.F.L., J.L.S. and C.H. analysed and interpreted data. A.F.F., K.D., K.S., Bv.L., J.H.S. and C.H. wrote the manuscript. T.U., J.J.N., G.D.P., J.W.L., K.M.M., and J.L.S. edited the manuscript. All authors read and approved the final version.

## Funding

## Competing interests

J.H.S., Bv.L., S.S.D., J.W.L., G.D.P. and K.M.M. are full-time employees and shareholders of the Denali Therapeutics Inc. CH collaborates with Denali Therapeutics and is a member of the advisory boards of AviadoBio and Cure Ventures. MBr is a member of the Neuroimaging Committee of the EANM. MB has received speaker honoraria from Roche, GE Healthcare, Iba, Miltenyi, and Life Molecular Imaging; has advised Life Molecular Imaging, MIAC, Cenos, and GE Healthcare; and is currently on the advisory or imaging review boards of AC Immune and ZRO Imaging. The remaining authors declare no competing interests.

## Additional information

[1]Metabolic Biochemistry, Biomedical Center, Faculty of Medicine, Ludwig-Maximilians Universität München, Munich, Germany. [2]German Center for Neuro-
degenerative Diseases (DZNE), Munich, Germany. [3]German Center for Neurodegenerative Diseases (DZNE), Bonn, Germany. [4]Department of Pediatrics,
University Hospital Würzburg, Würzburg, Germany. [5]Genomics and Immunoregulation, Life & Medical Sciences (LIMES) Institute, University of Bonn,
Bonn, Germany. [6]Denali Therapeutics, Inc., South San Francisco, California 94080, USA. [7]Department of Nuclear Medicine, University Hospital, Ludwig-
Maximilians Universität München, Munich, Germany. [8]Platform for Single Cell Genomics and Epigenomics at the German Center for Neurodegenerative
Diseases (DZNE), University of Bonn, Western German Genome Center (WGGC), Bonn, Germany. [9]Biomedical Center, Faculty of Medicine, Core Facility Flow
Cytometry, Ludwig-Maximilians Universität München, Munich, Germany. [10]Neuroproteomics, School of Medicine and Health, TUM University Hospital,
Technical University of Munich (TUM), Munich, Germany. [11]Modular High-Performance Computing and Artificial Intelligence, Systems Medicine, German
Center for Neurodegenerative Diseases (DZNE), Bonn, Germany. [12]Institute for Stroke and Dementia Research,, University Hospital, Ludwig-Maximilians
Universität München, Munich, Germany. [13]Normandie University, UNICAEN, INSERM UMR-S U1237, Physiopathology and Imaging of Neurological Disorders
(PhIND), Caen, France. [14]Munich Cluster for Systems Neurology (SyNergy), Munich, Germany. [15]Institute of Neuronal Cell Biology, Technical University Munich
(TUM), Munich, Germany. [16]Institute of Developmental Genetics, Helmholtz Zentrum, Munich, Germany. [17]Technical University Munich (TUM), TUM Senior
Excellence Faculty (SEF), Munich, Germany. [18]These authors contributed equally: Astrid F. Feiten, Kilian Dahm, Kai Schlepckow. [19]These authors jointly
supervised this work: Kathryn M Monroe, Joachim L Schultze, Christian Haass ✉e-mail: monroe@dnli.com; joachim.schultze@dzne.de; chris-
tian.haass@dzne.de

