## [Transparent Peer Review file · Nature Communications]

TREM2 expression level is critical for microglial state, metabolic capacity and efficacy of TREM2 agonism

Corresponding Author: Professor Christian Haass

Version 0:

Reviewer comments:

Reviewer #1

(Remarks to the Author)

The manuscript entitled “TREM2 expression level is critical for microglial state, metabolic capacity and efficacy of TREM2 agonism” from Feiten and Dahm, et al. report the creation of a novel TREM2 reporter mouse model that reveals a rich heterogeneity of TREM2-dependent microglial transcriptomic and metabolic states in mouse models of Alzheimer’s disease (AD). Specifically, they show a clear upregulation of TREM2 levels in microglia around A β plaques, and recapitulate the disease-associated microglia (DAM) transcriptomic signature. With their model, the authors are among the first to distinguish the independent impacts of A β pathology and TREM2 levels on microglial metabolism. Importantly, this study shows that microglial metabolic changes in response to TREM2 agonism are also critically modulated by TREM2 expression levels.

Overall, this is an important and timely study that provides important insights into microglial heterogeneity with implications for developing microglia-targeting therapies in AD. It generates a useful TREM2 reporter and rich, high-quality microglia metabolic datasets for the field. The authors may consider the following points helpful to improve an already strong manuscript:

Comments:

1. The authors provide a wealth of microglia metabolic data and its association with TREM2. Data showing how the TREM2-dependent microglia metabolic shift impacts their phenotype in response to A β (i.e., survival, proliferation, phagocytosis, degradation) should be provided.

2. A number of studies to date have described differences between human and mouse TREM2 and humanized TREM2 mouse models have been developed to address the issue (PMC7584603, PMC5839761). To bolster this study’s translational implications, can the authors provide evidence for a gradient of microglial TREM2 expression and metabolic dysfunction around A β plaques in human postmortem brain samples?

3. In other settings, it has been shown that targeting a cell surface protein can result in changes in abundance—for example, targeting TFRC receptor on brain endothelial cells can drive its downregulation/degradation (PMC3920563). Can the authors provide data on how TREM2 agonists may alter TREM2 levels? This would be particularly important if such changes create a mismatch between TREM2 and mKate levels and shape interpretation of Figure 6.

Minor comments:

1. Anti-amyloid monoclonal antibodies (mAbs) have been shown to reduce the amyloid plaque burden and there is significant clinical interest in combination therapies. Could the authors investigate or at least discuss potential microglial metabolic changes induced by their TREM2 agonists when given in combination with anti-A β mAbs such as Lecanemab?

2. Have the authors examined mKate2 levels in TREM2-expressing CNS-residing or infiltrating myeloid cells other than microglia? If so, please, provide evidence or discuss.

3. Please, provide scale bars in Fig. 1B and top panel of Fig. 1D

4. As reviewed in Qu and Li, 2024, microglial TREM2 expression appears to be age and brain-region dependent, and may vary between humans and mice. Please, clarify which brain regions have been examined for mKate2 expression in Fig. 1D and 1F, and share reasoning for the age of animals in transcriptomic and metabolic profiling.

Reviewer #2

(Remarks to the Author)

The manuscript by Feiten et al., entitled "TREM2 expression level is critical for microglia state, metabolic capacity and efficacy of TREM2 agonism" discusses the role of TREM2 (Triggering Receptor Expressed on Myeloid Cells 2) expression levels in regulating microglial function, metabolism, and its potential therapeutic targeting in neurodegenerative diseases like Alzheimer's disease. It provides a detailed study using a novel TREM2 reporter mouse model generated by the authors to explore how varying levels of TREM2 expression in microglia influence metabolic pathways, especially glucose metabolism and cholesterol processing and handling.

Key points:

1. TREM2 in Alzheimer's disease (AD):

TREM2 has been identified as a key player in microglial activation and in mediating immune responses in the brain, especially in the context of AD pathology. Variants of TREM2 are associated with increased risk for late-onset AD, and the paper aims to better understand the molecular and functional changes that occur in microglia when TREM2 is either upregulated or downregulated.

2. Methodology:

The paper used a novel TREM2 reporter mouse model generated and validated by the authors to study TREM2's role in microglia.

The researchers isolated microglia based on TREM2 expression levels and performed RNA sequencing, metabolomic and lipidomic analyses to identify changes in metabolic pathways.

The study examined microglial response both in healthy mice and those with amyloid plaque pathology (using the APP/PS1 mouse model of AD).

3. Main Findings:

Plaque proximity and TREM2 expression: Microglia in closer proximity to amyloid plaques showed increased expression of TREM2. The study highlights that TREM2 is not uniformly expressed but varies depending on the microglial environment and plaque burden.

Metabolic changes: Microglia with high TREM2 levels exhibited increased oxidative phosphorylation, cholesterol metabolism, and glucose uptake. This aligns with previous studies that link TREM2 activation with microglial survival and metabolic fitness.

TREM2 agonists: The paper also explores the use of a TREM2-activating antibody and identifies a window of TREM2 expression where microglia respond optimally to treatment. This finding is important for designing future clinical trials targeting TREM2 in AD.

4. Clinical implications:

The authors suggest that understanding the levels of TREM2 expression in microglia is essential for the effective application of TREM2-targeting therapies. Too little or too much TREM2 expression might reduce the therapeutic effect, emphasizing the need for precise timing and dosage in potential treatments. Moreover, the authors also suggest that TREM2 (loss of function) risk variants are likely associated with reduced cerebral glucose metabolism and regional hypometabolism, which could serve as a novel biomarker of prodromal AD in patients.

5. Strengths:

The paper provides a robust and detailed methodology, using state-of-the-art techniques such as CRISPR/Cas9 editing, RNA-sequencing, and metabolomic profiling to dissect TREM2's role in microglial metabolism.

The research connects basic molecular findings with clinical applications, providing a strong rationale for targeting TREM2 in therapeutic interventions for AD.

6. Areas for further exploration:

The study primarily focuses on microglial metabolism and its interaction with amyloid plaques. It would be interesting to see more detailed work on how these metabolic changes directly influence neuronal health and cognitive functions. The paper touches on the potential for TREM2 as a biomarker (soluble TREM2 in cerebrospinal fluid) as well as hypometabolic states as prodromal AD markers but doesn't go into much depth on this. Further exploration in clinical

settings would be valuable.

Overall, the paper makes a significant contribution to understanding how microglial metabolic changes, regulated by TREM2 expression levels, are central to their function in disease contexts like Alzheimer's. This insight is critical as the field moves toward TREM2-targeting therapies and I highly recommend the publication of this manuscript in Nature Communications.

Reviewer #3

(Remarks to the Author)

In this manuscript by Feiten and Dahm et al. the authors have developed a novel reporter mouse model. This reporter consists of an ER-bound fluorescent protein that is co-expressed with TREM2 from the endogenous TREM2 locus. The authors show that the fluorescence of this mKate2-KDEL reporter increases after traumatic brain injury, as does TREM2 RNA. They also show higher mKate2-KDEL around plaques. Subsequently they use this reporter to sort -and analyze- microglial populations with low- mid- and high TREM2 reporter expression. Based on the presence or absence of TREM2 they define TREM2-dependent and independent changes in microglial state. Lastly, using this approach they show that the effect of a TREM2 agonist on microglial lipidomes is stronger in TREM2-high expressing microglia.

Major points

- In general, the conclusions support existing literature, but few novel findings are being presented, besides the TREM2 medium population responding best to the TREM2 agonizing antibody.
- For figures 1-4, it is unclear what the benefit of the reporter mouse is over using TREM2 surface staining for cell sorting, or using scRNA-seq with TREM2 mRNA as internal reference for TREM2-associated DEGs. For the latter, several datasets exist and could be used for comparison. It is also unclear how dynamic the reporter is (what is the turnover time, how does this relate to mRNA changes)
- Throughout the manuscript, for example line 36 in the abstract, TREM2 is claimed to be causal for changes in gene expression (TREM2-dependent and independent effects), but in most cases the manuscript only shows association, not causality.
- Is there a functional difference in TREM2 med and TREM2 high microglia after TREM2 ATV treatment? Do the TREM2 med cells indeed become more phagocytic, in contrast to the TREM2 high cells? These experiments would provide a more solid basis to analyse differential effects of TREM2 agonizing antibodies.

Minor points

- The RNA-seq analyses are not well described or it is not clear why this analysis is chosen (such as Figure 3G-3J, why is this done instead of a general pathway level or GO-term analysis)
- In the first experiments most microglia are in the TREM2 med/high range (Fig. 2B), while in the experiments with the TREM2 ATV, most cells are in the TREM2 low range (Ext. Data 4C). Is this a difference in mouse model? And what implications does this have for the conclusions?
- Do the fractions of TREM2 med and TREM2 high cells change after antibody treatment (related to Ext. Data 4C)?
- In line 271, extended data 5D should be 4D.
- Are TREM2 low cells low in TREM2 or negative for TREM2? In other words, does an isotype control provide different labeling or is it similar?

Reviewer #4

(Remarks to the Author)

Version 1:

Reviewer comments:

Reviewer #1

(Remarks to the Author)

The authors have thoroughly addressed my comments. Congratulations on an important study.

Reviewer #2

(Remarks to the Author)

The authors addressed all comments and added significant data to strengthen their observations. I have no reservations about the relevancy of the study.

Reviewer #3

(Remarks to the Author)

All my concerns have been addressed and I congratulate the authors on their very interesting and important study.

Reviewer #4

(Remarks to the Author)

REVIEWER COMMENTS

Reviewer #1 (Remarks to the Author):

The manuscript entitled “TREM2 expression level is critical for microglial state, metabolic capacity and efficacy of TREM2 agonism” from Feiten and Dahm, et al. report the creation of a novel TREM2 reporter mouse model that reveals a rich heterogeneity of TREM2-dependent microglial transcriptomic and metabolic states in mouse models of Alzheimer’s disease (AD). Specifically, they show a clear upregulation of TREM2 levels in microglia around A β plaques, and recapitulate the disease-associated microglia (DAM) transcriptomic signature. With their model, the authors are among the first to distinguish the independent impacts of A β pathology and TREM2 levels on microglial metabolism. Importantly, this study shows that microglial metabolic changes in response to TREM2 agonism are also critically modulated by TREM2 expression levels.

Overall, this is an important and timely study that provides important insights into microglial heterogeneity with implications for developing microglia-targeting therapies in AD. It generates a useful TREM2 reporter and rich, high-quality microglia metabolic datasets for the field. The authors may consider the following points helpful to improve an already strong manuscript:

Comments:

1. The authors provide a wealth of microglia metabolic data and its association with TREM2. Data showing how the TREM2-dependent microglia metabolic shift impacts their phenotype in response to A β (i.e., survival, proliferation, phagocytosis, degradation) should be provided.

We followed the suggestion of the reviewer and investigated the functional consequences of the TREM2-dependent microglial metabolic shift by investigating phagocytosis as recommended. To do so, we added pHrodo-labeled myelin to freshly prepared brain single-cell suspensions from 9-10-months-old Trem2-mKate2.APP/PS1 mice to examine myelin phagocytosis by microglia. Using FACS, we quantified both the percentage of pHrodo-positive cells and the median fluorescence intensity in the low, mid, and high mKate2 subpopulations. We found that microglia gradually increase their phagocytic capacity with increasing mKate2 expression (**new Figure 6 and new Supplementary Figure 4, page 11/12 of revised manuscript**). This finding is in line with the gradual increase in metabolic capacity shown in our transcriptomic (Fig. 3G-J), glucose uptake (Fig. 4) and metabolomic analyses (Fig. 5). Furthermore, we also conducted the myelin phagocytosis assay with microglia from a second

independent amyloid mouse model expressing endogenous APP (Trem2-mKate2.App^{SAA}). Again, we detected a gradual increase of the phagocytic capacity of microglia with increasing mKate2 expression validating our findings from Trem2-mKate2.APP/PS1 mice. These findings are now also shown in the new Fig. 6 and new Supplementary Figure 4 (page of 11/12 of revised manuscript).

New Figure 6: Gradual increase in microglial metabolic capacity correlates with gradual increase in phagocytic capacity.

New Supplementary Figure 4: FACS analysis of myelin phagocytosis by mKate2 low, mid, and high microglia in two independent mouse models of amyloid pathology.

2. A number of studies to date have described differences between human and mouse TREM2 and humanized TREM2 mouse models have been developed to address the issue (PMC7584603: McQuade et al, 2020; PMC5839761: Song et al, 2018). To bolster this study's translational implications, can the authors provide evidence for a gradient of microglial TREM2 expression and metabolic dysfunction around A β plaques in human postmortem brain samples?

Although the TREM2 signaling pathway is well conserved between mouse and humans, murine and human microglial populations show clear gene expression differences, whose functional consequences are not understood. To provide evidence for a gradient of TREM2 expression around human amyloid plaques, human postmortem single-nucleus (sn) RNA-seq data of the dorsolateral prefrontal cortex and middle temporal gyrus published by Gabitto et al (2024) (PMID 39402379)

were examined filtering for patients with no or high Alzheimer's disease (AD) neuropathologic change (ADNC), retrieving a total of 46,345 microglia (**new Supplementary Response Letter Figure E, page 8 of revised manuscript**). The plaque proximity of TREM2-expressing microglia was estimated by evaluating the enrichment of the plaque-induced gene (PIG) signature by Chen et al (2020) (PMID 32702314) against the respective TREM2 expression (**new Supplementary Response Letter Figure F, page 8 of revised manuscript**). In both, non-AD patients and patients with high ADNC a moderate correlation between the TREM2 expression and the inferred plaque proximity was observed. Furthermore, a significant difference between both patient groups was measured suggesting an increased TREM2 expression in patients with high ADNC based on the plaque proximity.

New Supplementary Figure 2E,F: Increased TREM2 expression in patients with high ADNC based on the plaque proximity.

Next, we investigated if we could detect a metabolic shift in human postmortem microglia. In this case, we modeled the inferred plaque proximity against the enrichment scores of cholesterol homeostasis genes enriched in the steelblue network module (Fig. 3H, **Response Letter Response Letter Figure**). Indeed, the expression of cholesterol homeostasis genes increased in patients with high ADNC based on the inferred plaque proximity, while this was not the case in non-AD patients. Comparing a null linear mixed-effect model to a model split by the ADNC by means of a likelihood ratio test indicates statistical differences just above the $p < 0.05$ threshold between the two models. Taken together, by using snRNA-seq of human postmortem microglia we detected an increase in TREM2 expression in patients with high ADNC based on the plaque proximity (**new Supplementary**

Figure 2F, page 8 of revised manuscript) while the expression of cholesterol homeostasis genes exhibited a strong trend toward an increase as well (Response Letter Figure 1).

Response Letter Figure 1: Trend of increased cholesterol homeostasis gene signatures according to plaque proximity in postmortem microglia of patients with high ADNC.

Linear regression analysis of the enrichment score of the plaque-induced gene (PIG) signature against the enrichment scores of genes part of the Molecular Signature Database (MSigDB) Hallmark geneset 'Cholesterol homeostasis' enriched in the steelblue network module (Fig. 3H) in TREM2-expressing microglia from Gabitto et al. (2024) (PMID 39402379). Dots and trend lines are colored based on the AD neuropathologic change (ADNC). Pearson's correlation coefficients and p-values are documented for each trendline and the p-value of a likelihood ratio test (LRT) comparing a null linear mixed-effect model to a model split by the ADNC is indicated.

3. In other settings, it has been shown that targeting a cell surface protein can result in changes in abundance—for example, targeting TFRC receptor on brain endothelial cells can drive its downregulation/degradation (PMC3920563: Bien-Ly et al (2014), J Exp Med). Can the authors provide data on how TREM2 agonists may alter TREM2 levels? This would be particularly important if such changes create a mismatch between TREM2 and mKate levels and shape interpretation of Figure 6.

We agree with this reviewer that TREM2 levels may be altered upon engagement with an anti-TREM2 antibody. This was indeed shown in a previous study where a similar ATV-enabled antibody led to a decrease in surface TREM2 likely as a result of TfR-mediated receptor internalization, but not *total* levels in HEK293 cells overexpressing TREM2 upon 10 minutes of treatment (see in Van Lengerich et al (2023) Extended Data Fig. 4H, PMID 36635496). However, in our experiment, we collected mice one month after the last dose of the TREM2 agonist, i.e., ATV:4D9. Therefore, it is unlikely that TREM2 levels will be different between ATV:ISO and ATV:4D9 dosed mice since at this timepoint the antibody has long been cleared from tissue and circulation. Nevertheless, we generated whole brain RIPA lysates

from snap-frozen brain hemispheres we had collected in an independent experiment which was run in an identical manner to the experiment described in the **new Figure 7 (i.e. former Fig. 6)**. Mass spectrometry based proteomic analyses confirmed that TREM2 levels in whole brain RIPA lysates were unaltered upon ATV:4D9 treatment at this timepoint (see **new Supplementary Fig. 5E and Fig. 5F, page 13 of revised manuscript**). The proteomic dataset has been deposited to the ProteomeXchange Consortium via the PRIDE partner repository.

New Supplementary Figure 5E,F: Unaltered TREM2 levels in whole brain RIPA lysates upon ATV:4D9 treatment as determined by mass spectrometry.

In addition to TREM2 protein quantification, we also conducted transcriptomic analyses of the TREM2 low, mid, and high subpopulations from the same mice which were used to generate the lipidomic dataset shown in the **new Fig. 7 (i.e. former Fig. 6)**. This revealed no downregulation of expression of key DAM genes such as *Trem2* and *ApoE* upon chronic ATV:4D9 treatment across all subpopulations (see **new Supplementary Fig 5D, page 13 of revised manuscript**); these data are consistent with our observation that TREM2 protein levels are not altered one month after the final antibody treatment (see above). Importantly, we note that the gradual upregulation of key DAM genes in the TREM2 low, mid and high subpopulations in *App^{SAA}* mice is very similar to what we had seen in *APP/PS1* mice (see Fig. 2B and Fig. 3E,F) strongly suggesting that our reporter essentially captures the same transcriptomic alterations in subpopulations of microglia in two independent mouse models of amyloid pathology.

New Supplementary Figure 5D: mRNA levels of key DAM genes are unaltered in microglial subpopulations upon ATV:4D9 treatment.

Minor comments:

1. Anti-amyloid monoclonal antibodies (mAbs) have been shown to reduce the amyloid plaque burden and there is significant clinical interest in combination therapies. Could the authors investigate or at least discuss potential microglial metabolic changes induced by their TREM2 agonists when given in combination with anti-A β mAbs such as Lecanemab?

We absolutely agree with this reviewer that there is an urgent need to actively explore combination therapies. However, investigating potential microglial metabolic changes induced by anti-TREM2 antibodies given in combination with anti-A β antibodies essentially constitutes an entire study and line of investigation on its own and is thus beyond the scope of the current manuscript. Nevertheless, we now discuss the importance of combination treatments and potential microglial metabolic changes in the Discussion section (**page 17 of revised manuscript**).

2. Have the authors examined mKate2 levels in TREM2-expressing CNS-residing or infiltrating myeloid cells other than microglia? If so, please, provide evidence or discuss.

We thank the reviewer for raising this important point. Indeed, microglia are not the only cells in the CNS expressing TREM2 since its expression in border-associated macrophages, such as perivascular macrophages, has been documented and is known to increase during AD progression in humans (see Yang et al (2022), PMID 35165441). Since we used whole brain tissue and thus did not specifically isolate these cells we cannot comment on their mKate2 expression levels.

3. Please, provide scale bars in Fig. 1B and top panel of Fig. 1D

We apologize for the absence of scale bars in Fig. 1B and top panel of the **new Fig. 1F (i.e. former Fig. 1D)**. We have now added scalebars accordingly.

4. As reviewed in Qu and Li, 2024 (PMID: 34470515: „Microglial TREM2 at the intersection of brain aging and Alzheimer’s disease“), microglial TREM2 expression appears to be age and brain-region dependent, and may vary between humans and mice. Please, clarify which brain regions have been examined for mKate2 expression in Fig. 1D and 1F, and share reasoning for the age of animals in transcriptomic and metabolic profiling.

For the data shown in the **new Fig. 1F (i.e. former Fig. 1D)**, we examined cortical sections of Trem2-mKate2 animals (we now added this information to the Figure legend).

In the **new Fig. 1H (i.e. former Fig. 1F)** we examined hippocampal sections of Trem2-mKate2.*APP/PS1* mice at an age of 4 months (we now added this information to the Figure legend). We chose the hippocampus and an age of 4 months to have a rather low level of amyloid plaque pathology since widespread plaque pathology would make it very difficult to identify microglia in proximity of individual plaques (or even in plaque-free areas).

Reasoning for the age of animals in transcriptomic and metabolic profiling:

In Figs. 2, 3 and 5 we report transcriptomic as well as metabolomic data which were collected from 9-month-old *APP/PS1* mice. Since this mouse model is an aggressive APP-overexpressing model animals at this age are quite advanced in terms of disease progression, i.e., they exhibit strong activation of the DAM phenotype characterized by a substantial increase in TREM2 expression. We therefore chose mice at this age to increase the chance to capture different microglial states according to differential mKate2 expression. In the **new Fig. 6** and **new Fig. 7 (i.e. former Fig. 6)**, however, we employed the *App*^{SAA} mouse model. Since the *App*^{SAA} mouse model is an *App* knock-in model characterized by slower disease progression compared to the *APP/PS1* mouse model (Xia et al (2022), PMID 35690868) we designed the ATV:4D9 treatment study such that mice were collected at an older age, i.e. at 12 months.

Reviewer #2 (Remarks to the Author):

The manuscript by Feiten et al., entitled “TREM2 expression level is critical for microglia state, metabolic capacity and efficacy of TREM2 agonism” discusses the role of TREM2 (Triggering Receptor Expressed on Myeloid Cells 2) expression levels in regulating microglial function, metabolism, and its potential therapeutic targeting in neurodegenerative diseases like Alzheimer's disease. It provides a detailed study using a novel TREM2 reporter mouse model generated by the authors to explore how varying levels of TREM2 expression in microglia influence metabolic pathways, especially glucose metabolism and cholesterol processing and handling.

Key points:

1. TREM2 in Alzheimer's disease (AD):

TREM2 has been identified as a key player in microglial activation and in mediating immune responses in the brain, especially in the context of AD pathology. Variants of TREM2 are associated with increased risk for late-onset AD, and the paper aims to better understand the molecular and functional changes that occur in microglia when TREM2 is either upregulated or downregulated.

2. Methodology:

The paper used a novel TREM2 reporter mouse model generated and validated by the authors to study TREM2's role in microglia.

The researchers isolated microglia based on TREM2 expression levels and performed RNA sequencing, metabolomic and lipidomic analyses to identify changes in metabolic pathways. The study examined microglial response both in healthy mice and those with amyloid plaque pathology (using the APP/PS1 mouse model of AD).

3. Main Findings:

Plaque proximity and TREM2 expression: Microglia in closer proximity to amyloid plaques showed increased expression of TREM2. The study highlights that TREM2 is not uniformly expressed but varies depending on the microglial environment and plaque burden.

Metabolic changes: Microglia with high TREM2 levels exhibited increased oxidative phosphorylation,

cholesterol metabolism, and glucose uptake. This aligns with previous studies that link TREM2 activation with microglial survival and metabolic fitness.

TREM2 agonists: The paper also explores the use of a TREM2-activating antibody and identifies a window of TREM2 expression where microglia respond optimally to treatment. This finding is important for designing future clinical trials targeting TREM2 in AD.

4. Clinical implications:

The authors suggest that understanding the levels of TREM2 expression in microglia is essential for the effective application of TREM2-targeting therapies. Too little or too much TREM2 expression might reduce the therapeutic effect, emphasizing the need for precise timing and dosage in potential treatments. Moreover, the authors also suggest that TREM2 (loss of function) risk variants are likely associated with reduced cerebral glucose metabolism and regional hypometabolism, which could serve as a novel biomarker of prodromal AD in patients.

5. Strengths:

The paper provides a robust and detailed methodology, using state-of-the-art techniques such as CRISPR/Cas9 editing, RNA-sequencing, and metabolomic profiling to dissect TREM2's role in microglial metabolism.

The research connects basic molecular findings with clinical applications, providing a strong rationale for targeting TREM2 in therapeutic interventions for AD.

6. Areas for further exploration:

The study primarily focuses on microglial metabolism and its interaction with amyloid plaques. It would be interesting to see more detailed work on how these metabolic changes directly influence neuronal health and cognitive functions.

We absolutely agree with this reviewer. From a therapeutic point of view, it is crucial to investigate in detail whether TREM2 agonism may lead to ameliorated neuronal damage and improved cognitive functions. However, since mouse models of amyloid pathology as employed in our study are good models for amyloidogenesis but insufficient to study neurodegeneration and cognitive deficits it was not possible to address these important points in a scientifically appropriate manner.

The paper touches on the potential for TREM2 as a biomarker (soluble TREM2 in cerebrospinal fluid) as well as hypometabolic states as prodromal AD markers but doesn't go into much depth on this. Further exploration in clinical settings would be valuable.

We agree that CSF sTREM2 is an important biomarker to monitor microglial activation along the trajectory of AD progression. Please note that we (Suarez-Calvet et al (2016) PMID 27974666; Ewers et al (2019) PMID 31462511; Morenas-Rodriguez et al (2022) PMID 35305339) and others (Henjum et al (2016) PMID 27121148) already published data indicating that CSF sTREM2 may be used as a biomarker for microglial activation based on analyses from humans. In particular, we have shown in the DIAN cohort that a higher presymptomatic rate of sTREM2 change in the CSF correlates with a slower rate of cognitive decline in carriers of APP and PS1/PS2 mutations, which argues for a TREM2-dependent protective effect at an early disease stage (Morenas-Rodriguez et al (2022), PMID 35305339). Additionally, we have demonstrated that CSF sTREM2 levels may be used as a readout for target engagement upon dosing with anti-TREM2 antibodies (Schlepckow et al (2020), PMID 32154671; Van Lengerich et al (2023), PMID 36635496). Specifically, we have shown that CSF sTREM2 levels are reduced after 24 hours and 96 hours upon administration of a single dose of the mouse-specific antibody ATV:4D9 (Supplementary Fig. 1L in Van Lengerich et al (2023), PMID 36635496). Since this is the exact same antibody which was used in the current study we generated a CSF proteomic dataset to quantify CSF sTREM2 levels in ATV:ISO and ATV:4D9 groups. We found no alteration in CSF sTREM2 levels upon ATV:4D9 treatment which we expected since CSF was collected one month after the final dose, therefore, antibody-mediated changes to sTREM2 will be no longer observed (see **new Supplementary Fig. 5G and 5H, page 13 of revised manuscript**). The CSF proteomic dataset has been deposited to the ProteomeXchange Consortium via the PRIDE partner repository.

New Supplementary Figure 5G,H: Unaltered sTREM2 levels in CSF upon ATV:4D9 treatment as determined by mass spectrometry.

We also agree that hypometabolic states may serve as prodromal AD biomarkers. However, since amyloid mouse models show hypermetabolism instead of hypometabolism (Brendel et al (2016) PMID; 26912428; Li et al (2016) PMID 27763550; Xiang et al (2021) PMID 34644146) this point cannot be properly addressed within the scope of the current study based on a lack of translation modeling of this disease phenotype in the mouse.

Overall, the paper makes a significant contribution to understanding how microglial metabolic changes, regulated by TREM2 expression levels, are central to their function in disease contexts like Alzheimer's. This insight is critical as the field moves toward TREM2-targeting therapies and I highly recommend the publication of this manuscript in Nature Communications.

Reviewer #3 (Remarks to the Author):

In this manuscript by Feiten and Dahm et al. the authors have developed a novel reporter mouse model. This reporter consists of an ER-bound fluorescent protein that is co-expressed with TREM2 from the endogenous TREM2 locus. The authors show that the fluorescence of this mKate2-KDEL reporter increases after traumatic brain injury, as does TREM2 RNA. They also show higher mKate2-KDEL around plaques. Subsequently they use this reporter to sort -and analyze- microglial populations with low- mid- and high TREM2 reporter expression. Based on the presence or absence of TREM2 they define TREM2-dependent and independent changes in microglial state. Lastly, using this approach they show that the effect of a TREM2 agonist on microglial lipidomes is stronger in TREM2-high expressing microglia.

Major points

- In general, the conclusions support existing literature, but few novel findings are being presented, besides the TREM2 medium population responding best to the TREM2 agonizing antibody.

Reviewer #1 states that “this is an important and timely study that provides important insights into microglial heterogeneity with implications for developing microglia-targeting therapies in AD. It generates a useful TREM2 reporter and rich, high-quality microglia metabolic datasets for the field.” In addition, reviewer #2 concludes that “the paper makes a significant contribution to understanding how microglial metabolic changes, regulated by TREM2 expression levels, are central to their function

in disease contexts like Alzheimer's. This insight is critical as the field moves toward TREM2-targeting therapies and I highly recommend the publication of this manuscript in Nature Communications.”

In line with reviewers #1 and #2, we are convinced that the TREM2 reporter mouse model which we describe and characterize in our manuscript is an important additional tool for the field and will be crucial for further understanding of TREM2 biology and pre-clinical development of TREM2-directed therapeutics.

This is a summary of the major novel findings presented in our manuscript:

- TREM2 expression is upregulated in a gradient based on proximity to amyloid plaques as shown in the **new Fig. 1I (i.e. former Fig. 1G)**. Previous work had suggested that TREM2 expression is only upregulated once microglia get in physical contact with amyloid plaques (Chen et al (2020), PMID 32702314), suggesting a binary response that our findings refute.
 - We show that the amount of *Trem2* expressed by individual microglia is linked to that cells' metabolic state and capacity for glucose uptake as well as phagocytic uptake of myelin (see **new Fig. 6 and new Supplementary Fig. 4, page 11/12 of revised manuscript**). These findings link transcriptomics to metabolic activity and a key microglia function, which addresses an important outstanding gap in the field of how and whether transcriptional responses translate into microglia functions.
 - The TREM2 mid subpopulation is responding best to the TREM2 agonizing antibody as shown in the **new Fig. 7 (i.e. former Fig. 6)**, which is highly critical information given the number of TREM2 targeting therapies being currently examined in the clinic. Moreover, TREM2 high microglia respond in an opposite manner to the agonistic antibody, a finding, which is of greatest importance for all clinical efforts on TREM2 mediated stimulation of protective microglial functions. Thus, our data address a key outstanding question regarding how various heterogeneous microglial subpopulations respond to TREM2 agonism.
-
- For figures 1-4, it is unclear what the benefit of the reporter mouse is over using TREM2 surface staining for cell sorting, or using scRNA-seq with TREM2 mRNA as internal reference for TREM2-associated DEGs. For the latter, several datasets exist and could be used for comparison. It is also unclear how dynamic the reporter is (what is the turnover time, how does this relate to mRNA changes).

Several groups have encountered difficulties when staining surface TREM2 for cell sorting, possibly because of efficient shedding of TREM2 (Thornton et al (2017), PMID 28855301). Since the rapid turnover of TREM2 makes it difficult to identify microglia subpopulations just based on differences in TREM2 staining intensity, the retention of mKate2 within the ER enhances the chance to identify microglia subpopulations based on differential mKate2 signal intensity. Indeed, when we quantified TREM2 and mKate2 protein levels in HeLa cells upon transient transfection with the TREM2-P2A-mKate2 cDNA construct that was also used for generation of the novel reporter mouse model we found that TREM2 is turned over more rapidly than mKate2 due to its efficient and continuous shedding at the cell surface (see **new Fig. 1C and 1D, page 6 of revised manuscript**). That means that even in cells where cell surface TREM2 is lost due to the natural shedding process, our reporter still allows the detection of cells in which the TREM2 pathway was stimulated.

New Figure 1C,D: TREM2 is more rapidly degraded than mKate2 in HeLa cells upon transient transfection.

As we showed during the revision of our manuscript our reporter mouse model also enables the generation of functional data in a subpopulation-specific manner (see **new Fig. 6 and new Supplementary Fig.4, page 11/12 of revised manuscript**). The ability to acquire such data has been largely lacking in the field even though it is crucially needed for validating findings from transcriptomic analyses. More broadly, the TREM2 reporter mouse model will be a highly useful tool for modeling microglial responses to other microglial targeting therapeutics with improved resolution of subpopulations stratified by an important master regulator of microglial state and function, namely TREM2. Finally, it is a great resource for live imaging and related methods. Thus, we conclude that our

novel TREM2 reporter mouse model is an important methodological advancement that will better enable the field for basic research in addition to translational studies.

We also note that we compared our transcriptomic data from *APP/PS1* mice with RNA-seq datasets from Keren-Shaul et al (2017), PMID 28602351 and Grubman et al (2021), PMID 34021136 already in our manuscript during initial submission (see **new Supplementary Figs. 2G,H, i.e. former Supplementary Figs. 2F,G**) and found these to be highly consistent with the data reported in our manuscript.

- Throughout the manuscript, for example line 36 in the abstract, TREM2 is claimed to be causal for changes in gene expression (TREM2-dependent and independent effects), but in most cases the manuscript only shows association, not causality.

We agree with this statement and have accordingly toned down our wording at appropriate positions throughout the manuscript.

- Is there a functional difference in TREM2 med and TREM2 high microglia after TREM2 ATV treatment? Do the TREM2 med cells indeed become more phagocytic, in contrast to the TREM2 high cells? These experiments would provide a more solid basis to analyse differential effects of TREM2 agonizing antibodies.

Investigating the functional consequences of TREM2 agonism in the different subpopulations is indeed very informative. To this end, we carried out an ex vivo assay where we quantified both the percentage of pHrodo-positive cells and the median fluorescence intensity in the low, mid, and high mKate2 subpopulations upon phagocytosis of pHrodo-labeled myelin (see **new Fig. 6 and new Supplementary Fig. 4, page 11/12 of revised manuscript**). This revealed that microglia gradually increase their phagocytic capacity with increasing mKate2 expression in two independent mouse models of amyloid pathology.

This finding is in excellent agreement with our observations of increases in metabolic capacity (Fig. 3G-J), glucose uptake (Fig. 4), and abundance of key metabolites such as cholesterol, creatine and glutathione (Fig. 5). Importantly, we already demonstrated increased phagocytosis of pHrodo-labeled myelin by microglia isolated from mice which were *acutely* dosed with an ATV:TREM2 antibody (Van Lengerich et al (2023), PMID 36635496) suggesting agonism would also increase extent of phagocytosis among subpopulations. Of course, it would have been interesting to see how the *chronic* antibody treatment impacts on our observations as shown in the **new Fig. 6 and new Supplementary Fig. 4**

(page 11/12 of revised manuscript), however, we deemed *chronic* dosing of an additional cohort of 12-month-old mice beyond the scope of these revisions and hope that the reviewer agrees with this point.

New Figure 6: Gradual increase in microglial metabolic capacity correlates with gradual increase in phagocytic capacity.

New Supplementary Figure 4: FACS analysis of myelin phagocytosis by mKate2 low, mid, and high microglia in two independent mouse models of amyloid pathology.

Minor points

- The RNA-seq analyses are not well described or it is not clear why this analysis is chosen (such as Figure 3G-3J, why is this done instead of a general pathway level or GO-term analysis)

We thank the reviewer for this comment and incorporated a clearer description of the analysis conducted in the manuscript (page 9 of revised manuscript). A gene co-expression network analysis was chosen to characterize additional functional differences across the stratified microglia states at the same time (instead of two groups at a time in conventional pathway analysis) which were not covered by the AD- and *Trem2*-related DEGs. As the network is not restricted by p-value cut-offs, additional genes that are potentially not captured by the differential expression analysis are included in the network enabling the identification of novel gene modules and a more detailed functional

characterization of each microglia subpopulation. This type of analysis is well accepted within the transcriptomic field.

- In the first experiments most microglia are in the TREM2 med/high range (Fig. 2B), while in the experiments with the TREM2 ATV, most cells are in the TREM2 low range (Ext. Data 4C). Is this a difference in mouse model? And what implications does this have for the conclusions?

The distributions of mKate2 positive cells look indeed different between the two datasets. However, it is very likely that this observation is due to the inherent differences between the two amyloidosis mouse models employed in the current study. In Fig. 2A we use *APP/PS1* mice which are known to be among the most aggressive APP-overexpressing mouse models while in the **new Supplementary Fig. 5B (i.e. former Extended Data Fig.4C)** we use the *App^{SAA}* mouse model which is an *App* knock-in model known to be much less aggressive in terms of amyloid deposition (Xia et al (2022), PMID 35690868). Due to more advanced amyloid pathology in the *APP/PS1* model, microglial activation is more pronounced compared to the *App^{SAA}* model and the proportion of microglia in the mid/high mKate2 range is thus higher. The fact that the gradual increase in expression of key DAM markers in the low, mid, and high mKate2 subpopulations is very similar between these two independent mouse models of amyloid pathology (see Fig. 2B and Fig. 3E,F (*APP/PS1*) and **new Supplementary Fig. 5D, page 13 of revised manuscript (*App^{SAA})*) strongly suggests that our novel TREM2 reporter captures the same transcriptomic alterations in subpopulations of microglia from these two models due to the continuous buildup of amyloid pathology.**

New Supplementary Figure 5D: mRNA levels of key DAM genes in *App^{SAA}* mice gradually increase with increasing mKate2 expression very similar to what is observed in *APP/PS1* mice and are unaltered upon ATV:4D9 treatment.

- Do the fractions of TREM2 med and TREM2 high cells change after antibody treatment (related to Ext. Data 4C)?

We quantified the relative abundances of the TREM2 low, mid and high subpopulations from the number of cells sorted in the FACS analysis. This revealed no significant alteration between ATV:ISO and ATV:4D9 groups (**Response letter Figure 2**).

Response Letter Figure 2: Relative abundances of TREM2 low, mid and high subpopulations upon chronic ATV:4D9 treatment in *App*^{SAA} mice analyzed by two-way ANOVA.

- In line 271, extended data 5D should be 4D.

We thank this reviewer for bringing this mistake to our attention. We have corrected it at the respective position in the manuscript.

- Are TREM2 low cells low in TREM2 or negative for TREM2? In other words, does an isotype control provide different labeling or is it similar?

Since we subjected all of the subpopulations to RNA-seq we can confirm that the TREM2 low cells do in fact have low levels of *Trem2* mRNA and are not a non-expressing fraction (Fig. 2B). In addition, the metabolomic data as presented in the **new Fig. 7E (former Fig. 6E)** show that some metabolites are significantly changed in their abundance in the TREM2 low subpopulation upon ATV:4D9 treatment, which argues that TREM2 protein must be present on the cell surface since otherwise the antibody would not be able to modulate the metabolome in these cells.

Reviewer #4 (Remarks to the Author):
